# Thwaites Glacier thins and retreats fastest where ice-shelf channels intersect its grounding zone

Allison M. Chartrand[1,2], Ian M. Howat[3], Ian R. Joughin[4], Benjamin E. Smith[4]

[1]Earth System Science Interdisciplinary Center, University of Maryland, College Park, MD, USA
[2]NASA Goddard Space Flight Center, Greenbelt, MD, USA
[3]Byrd Polar and Climate Research Center, Ohio State University, Columbus, OH, USA
[4]Applied Physics Laboratory, University of Washington, Seattle, WA, USA

*Correspondence to*: Allison M. Chartrand (allison.chartrand@nasa.gov)

**Abstract.** Antarctic ice shelves buttress the flow of the ice sheet but are vulnerable to increased basal melting from contact with a warming ocean and increased mass loss from calving due to changing flow patterns. Channels and similar features at the bases of ice shelves have been linked to enhanced basal melting and observed to intersect the grounding zone, where the greatest melt rates are often observed. The ice shelf of Thwaites Glacier is especially vulnerable to basal melt and grounding-zone retreat because the glacier has a retrograde bed leading to a deep trough below the grounded ice sheet. We use digital surface models from 2010–2022 to investigate the evolution of its ice-shelf channels, grounding zone position, and the interactions between them. We find that the highest sustained rates of grounding-zone retreat (up to 0.7 km yr$^{-1}$) are associated with high basal melt rates (up to ~250 m yr$^{-1}$) and are found where ice-shelf channels intersect the grounding zone, especially atop steep local retrograde slopes where subglacial channel discharge is expected. We find no areas with sustained grounding zone advance, although some secular retreat was distal from ice-shelf channels. Pinpointing other locations with similar risk factors could focus assessments of vulnerability to grounding zone retreat.

## 1 Introduction

Thwaites Glacier in the Amundsen Sea region of West Antarctica has the potential to contribute up to 65 cm of sea-level rise (Rignot et al., 2019; Morlighem et al., 2020) and has experienced recent speed-up, ice shelf break-up, thinning, and grounding-line retreat (e.g. dos Santos et al., 2021). Thwaites Glacier lies atop an inland-sloping (retrograde) bed leading to a deep trough reaching 1.5 km below sea level (Morlighem et al., 2020). The glacier terminates in two distinct ice shelves, the Thwaites Eastern Ice Shelf (TEIS) and the Thwaites Western Ice Tongue (TWIT), collectively referred to as the Thwaites Glacier Ice Shelf (TGIS). Several studies have suggested that the Thwaites Glacier grounding zone (the region where the ice transitions from grounded to freely floating), may already be undergoing rapid, unstable retreat (Joughin et al., 2014; Goldberg et al., 2015; Rignot et al., 2014; Yu et al., 2018), in a process known as "marine ice sheet instability", or MISI (Gudmundsson et al.,

2012; Schoof, 2007; Weertman, 1974). Much of the basal melting and grounding zone retreat is attributed to contact with warm ocean water (Bevan et al., 2021; Schmidt et al., 2023; Holland et al., 2023) and loss of basal traction inland of the grounding zone (Joughin et al., 2024). Recent evidence suggests that tidal flexure allows seawater to intrude into the embayed grounding zone in the main trunk of the TWIT (located within Box C in Fig. 1), which may accelerate melting below intermittently grounded ice (Rignot et al., 2024). The TWIT has weakened rapidly over the past several decades (e.g. Miles et al., 2020), and the TEIS is expected to weaken significantly in the coming decades (e.g. Wild et al., 2022).

Several studies have shown that high rates of ice-shelf thinning and basal melting are often associated with ice-shelf channels, which are curvilinear incisions at the ice-shelf base believed to be maintained by buoyant meltwater plumes entrained within them along-flow (Alley et al., 2016; Drews, 2015; Chartrand and Howat, 2020; Gourmelen et al., 2017; Wearing et al., 2021). Others have shown that ice-shelf channels are also commonly associated with ice-shelf weakening through fracturing as a result of thinning (Vaughan et al., 2012; Dow et al., 2018; Alley et al., 2019). Ice-shelf channels initiate at the grounding zone and extend seaward. They often represent advected extensions of inverted troughs initiated by subglacial channelization beneath the grounded ice or incised by undulations in the bed (e.g. Le Brocq et al., 2013; Alley et al., 2016; Drews et al., 2017). Where subglacial channelization is present, the input of fresh subglacial meltwater may contribute to the growth of a buoyant meltwater plume that can entrain warm ocean water as it travels along the ice-shelf channel (Jenkins, 2011). However, it remains difficult to attribute the formation mechanism to any given channel, particularly if its surface expression does not intersect the grounding zone (e.g. Alley et al., 2016; Chartrand & Howat 2020). The TGIS has at least four previously mapped ice-shelf channels (Alley et al., 2016) and the grounded ice is underlain by at least two persistent subglacial channels intersecting the grounding zone (Hager et al., 2022). Recently, subglacial channels and aligned ice-shelf channels on Greenland ice tongues have been linked to thinning and retreat of the grounding line (Ciracì et al., 2023; Narkevic et al., 2023). It is unknown, however, the extent to which ice-shelf channels on the TGIS and/or subglacial channels within grounded ice may have contributed to the thinning and retreat at Thwaites Glacier.

In the absence of a method for directly measuring ice thickness from space, observations of ice-shelf channels and channel-like features (i.e. ice-shelf incisions oriented predominantly along-flow without clear evidence of subglacial initiation or entrained meltwater flow) and their relationship to variations in grounding zone position and other ice-shelf structures are only available from high-resolution (<100 m) measurements of surface height. While relatively frequent and accurate, observations from spaceborne altimetry, such as ICESat and ICESat–2, are limited to ground tracks. Recently, high-resolution digital surface models (DSMs) produced from stereoscopic satellite imagery, combined with altimetry, have been used to map changes in ice-shelf channels and other ice-shelf structures (Chartrand and Howat, 2020; Shean et al., 2019; Zinck et al., 2023). Using the extensive collection of repeat DSMs provided by the Reference Elevation Model of Antarctica (REMA) project (Howat et al., 2019), we map the positions of surface depressions overlying ice-shelf channels on the TGIS and subglacial channels within grounded ice as well as the landward extent of the transition to flotation as a proxy for the grounding line, termed the hydrostatic boundary (HB). We also construct the most comprehensive maps to date of time-evolving surface height, thickness, and basal mass change for the Thwaites Glacier and TGIS. Here we examine the transient locations of the

ice-shelf channels relative to those of the HB, as well as to variations in basal melt and flow speed, to assess the potential
relationship between channels and grounding-zone retreat.
**2 Datasets**
We use several geophysical datasets for this study as described throughout this section.
**2.1 REMA Digital Surface Model (DSM) strips and annual mosaics**
Reference Elevation Model of Antarctica (REMA) DSMs for the TGIS are obtained through stereophotogrammetry applied to
pairs of commercial submeter-resolution, panchromatic satellite images acquired by the MAXAR constellation, including
Worldview–1, –2 and –3, Quickbird–1 and –2, and GeoEye satellites (Howat et al., 2019). Elevations from REMA DSMs are
relative to the WGS84 ellipsoid and are distributed both as individual "strips" representing WGS84 ellipsoid elevations, created
from a single pair of images along their overlapping swath, and as seamless, continuous mosaics made from those strips. Both
product types are distributed at 2 m resolution, with downsampled versions available. We use the REMA 200 m mosaic for
the Thwaites region as a base map in several figures.
We utilise the DSM strips at 10 m resolution to map the HB (Section 3.1) and ice-shelf channels (Section 3.2). After
removing strips with insufficient coverage or low internal quality, as indicated in the metadata by a root-mean-squared error
value greater than 1 m, there are 191 strips for the TGIS.
We combine REMA strips into full-coverage (depending on strip availability) annual mosaics at 50 m resolution to
compute rates of change across the TGIS (Section 3.3). As strips are more readily available for summer months, REMA strips
from November to March are mosaicked to form an austral annual mosaic, for which we assign a nominal date of January 1.
For each annual mosaic, each strip is co-registered to every other using the method of Nuth and Kaab (2011). The co-registered
strips are then stacked, and the annual mosaic is the median height at each pixel through the stack (Fig. S1).
As part of this process, each strip and annual mosaic is registered to correct for single-value elevation biases using
ICESat–2, CryoSat–2, or Operation IceBridge (henceforth, IceBridge) altimetry data, depending on availability and/or which
dataset is temporally closest to the collection date of the strip. The registration dataset selected is based on a hierarchy; if the
first dataset is unavailable, the next dataset will be used, and so on. Priority is given to overlapping ICESat–2 ATL06 Version
5 (Smith et al., 2021) or Version 6 (Smith et al., 2023) ground control points (GCPs) that were collected within 10 days of the
strip or 100 days of the annual mosaic nominal date, then to GCPs from the ICEBridge Airborne Topographic Mapper (ATM)
L1B (Studinger, 2013), the IceBridge Land, Vegetation, and Ice Sensor (LVIS) L2 (Blair and Hofton, 2015), or the IceBridge
and ICECAP Riegl Laser Altimeter L2 (Blankenship et al., 2012) datasets collected within 10 or 100 days. Strips with no
contemporaneous ICESat–2 or IceBridge GCPs were registered to CryoSat–2 SARIn-mode elevations (European Space
Agency, 2023) collected within about 365 days of the strip. Unlike ICESat–2 and IceBridge registrations which register the
strip to temporally proximal GCPs, the CryoSat–2 registration is determined by a linear fit to elevation differences with respect
to time so that the DSM is fit to a temporal model of the elevation data. Strips that do not meet these registration criteria are
eliminated, leaving 177 strips over the TGIS (Fig. S2).
The strips and annual mosaics are not smoothed, but they are masked to remove artefacts and errors like clouds.
Following registration, the absolute residual, and mean and standard deviation thereof, between each DSM and the REMA
mosaic is computed. Then, the residual is smoothed by a 50 pixel (500 m for strips or 2500 m for annual mosaics) moving
mean, and any DSM pixels that correspond to a smoothed residual that is 2–5 standard deviations greater than the mean residual
are masked out; a larger magnitude in the residual mean or standard deviation triggers a smaller standard deviation threshold
for that strip. These thresholds were chosen to effectively remove artefacts and errors like clouds, but to maintain differences
due to the advection of surface features.
Following the procedure in Chartrand and Howat (2020), registered and masked strip and annual mosaic ellipsoid
elevations are converted to freeboard heights ($h$) by referencing to mean sea level using the EIGEN–6C4 geoid model (Förste
et al., 2014), correcting for Mean Dynamic Topography using the DTU22 MDT model (Knudsen et al., 2021), accounting for
firn density using the firn depth correction from Ligtenberg et al. (2011) provided in BedMachine, and correcting for tidal
variations using the CATS2008b inverse tide model (Padman et al., 2018).

## 2.2 BedMachine Antarctica

Bed heights referenced to the geoid are obtained from BedMachine Antarctica, Version 3 (Morlighem, 2022). The BedMachine
bed height and grounded ice thickness are used to estimate subglacial conditions (Section 3.4). This dataset also contains masks
for floating ice, grounded ice, ice-free land, and ocean which are used to select strips based on their overlapping area with the
relevant masks.

## 2.3 Ice velocities

Ice surface velocity for the TGIS is obtained from NASA Making Earth System Data Records for Use in Research
Environments (MEaSUREs) mosaicked, 450 m posting, InSAR–Based Antarctic Ice Velocity Map, Version 2 (henceforth,
"velocity mosaic"; Mouginot et al., 2012; Rignot et al., 2011, 2017), the MEaSUREs 1 km posting, SAR–Based Annual
Antarctic Ice Velocity Maps, Version 1 (henceforth, "annual velocity maps"; Mouginot et al., 2017a), and 250 m posting
velocity maps for the austral summer quarters (Oct–Dec and Jan–Mar) that we derived using speckle tracking applied to
Sentinel 1 A/B images from TGIS and Pine Island Glacier. The 450 m posting MEaSUREs mosaic is used to obtain masks for
ice moving $< 20$ m yr$^{-1}$ for registering strips and for filtering the annual and quarterly velocity maps. The MEaSUREs annual
velocity maps obtained for 2011–2015 are variable in their spatial coverage and quality, while the quarterly velocity maps
obtained for 2016–2023 have more consistent coverage and better quality. To obtain annual velocity maps from 2011–2023
with more consistent quality, we initially take different approaches to filling data gaps and reducing noise in each dataset: for
the 2011–2015 annual velocity maps, we take the average of each annual map and the velocity mosaic at each pixel; for the

2016–2023 quarterly maps, we take the average of each year's Oct–Dec map and Jan–Mar map at each pixel. Each resulting filled, noise-reduced annual map from the whole study period is then bilinearly interpolated to the same grid as the annual REMA mosaics and further filtered by taking the median velocity and standard deviation at each pixel throughout the time period and masking out pixels in each year where the velocity differs from the median pixel velocity by more than 2.5x the standard deviation for that pixel, then smoothing the velocity for each year with a 600 m moving mean window. The annual velocity maps are used to flow-shift the annual DSM mosaics to obtain Lagrangian rates of change and to investigate changes in velocity where we observe rapid HB retreat.

**2.4 Historical grounding lines**

The grounding line from the MEaSUREs Antarctic Boundaries for the 2007–2009 International Polar Year (IPY) from Satellite Radar, Version 2 dataset (Mouginot et al., 2017b) is used as a reference grounding line from which to measure changes in the grounding zone position and is henceforth termed 07–09 IPY GL or simply IPY GL. This dataset provides a complete and continuous grounding line derived from a variety of satellite platforms. Additional historical grounding lines are obtained for a long-term visual comparison (Fig. 1) from the MEaSUREs Antarctic Grounding Line from Differential Satellite Radar Interferometry, Version 2 for 7 February 1992 to 17 December 2014 (Rignot et al., 2016); however, these are not used for analyses.

**3 Methods**

We use a variety of previously defined and novel techniques to investigate changes on the TGIS, described below.

**3.1 Mapping the hydrostatic boundary**

The hydrostatic boundary (HB) is defined as the point at which the grounding thickness matches the flotation thickness. The grounding thickness is the distance between the observed ice surface and the bed from BedMachine. Flotation thickness ($H_E$) is determined from the DSM strip freeboard heights ($h$) as in Chartrand & Howat (2020, 2023):

$$H_E = h \frac{\rho_s}{\rho_s - \rho_i} - H_a \frac{\rho_a - \rho_i}{\rho_i - \rho_s}, \tag{1}$$

where $\rho_s$ is seawater density (1,027 kg m$^{-3}$), $\rho_i$ is meteoric ice density (918 kg m$^{-3}$), $\rho_a$ is the firn-air column density (2 kg m$^{-3}$), and $H_a$ is the thickness of the firn-air column within the freeboard (specifically, the length of the change in firn thickness resulting from compressing the firn column to ice density (Ligtenberg et al., 2011)). The subscript $E$ denotes that $H_E$ is an estimate of ice thickness.

We use the 07–09 IPY GL as the basis for where HBs are expected to be mapped. We track HBs at the continental grounding line and six pinning points (PP1–6) delineated in the IPY GL. For each strip, the difference between $H_E$ and the grounding thickness is computed at each pixel. This difference is converted to a contour map, and the coordinates of pixels

that lie on the 0 m contours are stored as features representing HBs near the continuous IPY GL and each pinning point. These
features are filtered and simplified for analysis as follows (Fig. S3). For each strip, HB features containing fewer than 25
coordinates are removed, and the remaining HB feature coordinates are smoothed by a 50 point moving mean (a distance of
about 200 m). If the strip has coverage over a given grounding line, a polygon is manually defined to encapsulate the HB
features that most likely represent that grounding line (i.e., the polygon is defined to keep the longest and most continuous HB
features and eliminate small isolated HBs), and points outside of the polygon are eliminated. Then, HB features are combined
by year, and an annual HB with a nominal date of January 1 is manually defined along the most inland features for the
continental GL, and the innermost features for each pinning point, from each year.

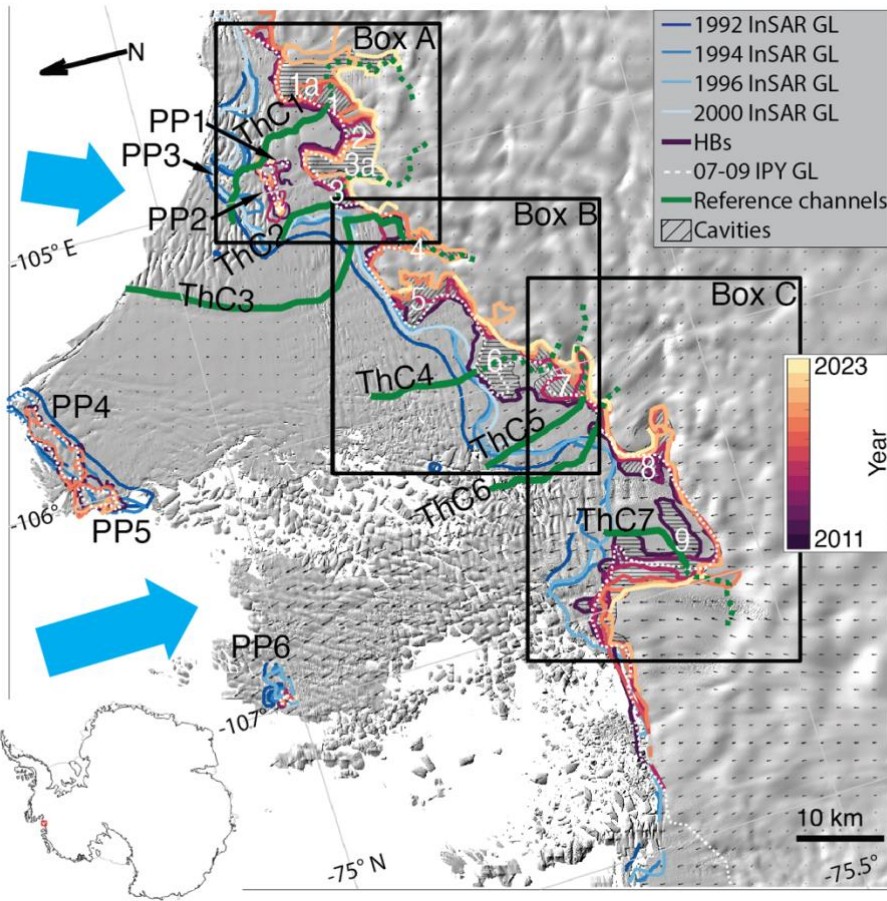


**Figure 1: REMA 200 m mosaic hillshade for the Thwaites Glacier and Thwaites Glacier ice shelf (TGIS) overlain by historical**
**grounding lines derived from InSAR (curves in shades of blue), the continuous InSAR-derived 07–09 IPY GL (white dashed curve,**
**including pinning points PP1–6), and selected hydrostatic boundaries (HBs) identified in this study (curves in shades of purple,**
**orange, and yellow, with lighter shades representing more recent HBs). The reference channels for the ice-shelf channels and surface**
**depressions identified in this study are shown as solid and dashed green curves, respectively, and labelled ThC1–7. Hatched regions**
**indicate cavities that opened as the HB retreated throughout the study period, labelled 1–9. Large blue arrows indicate the general**
location of circumpolar deep water (CDW) influx to the ice-shelf cavity (Dutrieux et al., 2014). The black boxes, Box A–C, indicate
the zoomed regions in following figures. Small black arrows represent ice flow.

## 3.2 Mapping ice-shelf channels and surface depressions

Complementary methods are used to locate persistent curvilinear ice-shelf basal features, including ice-shelf channels, as they are not directly observable using surface elevation alone. Sufficiently large ice-shelf channels (usually > 1 km wide) correspond to surface depressions that are resolvable as stream-like features in the surface topography (Drews, 2015). We do not attempt to verify the presence of entrained meltwater flow in the basal features we identify, and we refer to all consistently mapped basal incisions that are oriented predominantly parallel to flow and associated with surface depressions as ice-shelf channels. Ice-shelf channel locations from Alley et al. (2016) are used to initially query REMA DSM strips.

We map surface depressions over both grounded and floating ice by refining the method of using DSM local minima to map surface depressions (Chartrand and Howat, 2020). We compute maps of hypothetical stream channel depth for the ice-shelf surface from strip freeboard heights using the flow accumulation ("flowacc") function from the MATLAB-based TopoToolbox software (Schwangart & Scherler, 2014). We assume that features with high flow accumulation are surface depressions. We then compare potential depressions with DSM hillshade renderings, eliminating those that align with clear fractures (usually perpendicular to flow), are very short (< ~1 km), or do not intersect the ice shelf.

To identify ice-shelf channels on the shelf, we compute flotation thickness from strip freeboard height (Eq. 1) and determine the depth of the hydrostatic ice-shelf draft ($h - H_E$) relative to sea level. We invert the ice-shelf draft by multiplying its depth by –1, and again compute the hypothetical stream channel depth and flow accumulation across the inverted ice-shelf base, with the locations of stream flow identified as possible ice-shelf channels. As for the surface depressions, we remove spurious features by comparison with DSM hillshade renderings of both the surface and inverted basal topography.

Where available, potential ice-shelf channels are verified using IceBridge and pre-IceBridge MCoRDS L2 ice-penetrating radar (IPR) thicknesses (Paden et al., 2011, 2010) (Figs. S4–5). If a basal incision and/or surface depression does not match with a thickness minimum or a previously identified ice-shelf channel (e.g. Alley et al., 2016), we do not rule out that it could be an ice-shelf channel with entrained meltwater flow and look for other evidence of its formation.

## 3.3 Estimating rates of change

Time-evolving rates of change are estimated from the annual DSM mosaics within four epochs: 2011–2015, 2016–2019, 2020–2023, and the entire study period from 2011–2023. Within each epoch, rates of change are calculated from all combinations of annual mosaics such that the relevant quantity derived from the earlier mosaic in each combination is subtracted from the later mosaic and divided by the time elapsed between the mosaics. The Eulerian reference frame (fixed coordinate system, denoted by d$Q$/d$t$, where $Q$ is the quantity in question) is used over grounded ice, to prevent slope-induced errors, and the Lagrangian reference frame (coordinate system moves with ice flow, denoted by D$Q$/D$t$) is used over floating ice, where height

variability is dominated by horizontal advection. The strip-derived annual HB from each year is used to delineate the extent of floating and grounded ice for each annual mosaic. For grounded ice, we calculate the Eulerian rate of thickness change ($dH/dt$), where grounded ice thickness, $H$, is simply the DSM-derived surface height minus the BedMachine bed height. For floating ice, we calculate Lagrangian rates of ice-column surface height change ($Dh/Dt$), thickness change ($DH_E/Dt$), where flotation thickness $H_E$ is derived from annual DSM mosaic freeboard heights using Eq. 1, and basal mass loss or gain ($M_b$). For Lagrangian calculations, the mosaics are flow-shifted to a common date using the smoothed annual surface velocity maps (Section 2.3) following the approach of Shean et al. (2019) and Chartrand and Howat (2020). The mosaics are flow-shifted to 1 January of the earliest full year in each epoch (e.g., 1 January 2011 for the 2011–2015 epoch and the full study period). Lagrangian ice-column thinning can occur as a result of stretching as the ice accelerates (dynamic thinning) or as a result of surficial or basal ablation, although these mechanisms cannot be attributed by a calculation of $DH_E/Dt$, which only reflects how the surface height changes as the column advects due to the hydrostatic assumption.

The Lagrangian basal mass change rate $M_b$ (m ice equivalent yr$^{-1}$, negative values imply basal melt), for floating ice is determined from mass conservation as:

$$M_b = \frac{DH_E}{Dt} + H_E(\nabla \cdot \boldsymbol{u}) - M_s \qquad (2),$$

where $M_s$ is the surface accumulation rate (m yr$^{-1}$, positive for mass gain), and $\nabla \cdot \boldsymbol{u}$ is the divergence in the column-average horizontal velocity of the ice $\boldsymbol{u}$ (m yr$^{-1}$). As in Shean et al. (2019), the velocity divergence is computed at each time step prior to flow-shifting the DSM, so the $M_b$ estimate accounts for the flow history of each pixel. The rate of surface accumulation for Antarctica is obtained from the Regional Atmospheric Climate Model (RACMO) 3p2 (van Wessem et al., 2018) which provides estimates of $M_s$ for 1979–2016 on a 27 km grid. We bilinearly interpolate the per pixel mean $M_s$ from 2011–2016 to the mosaic grid coordinates and convert to ice-equivalent mass change rates. The maps of rates of change are smoothed by a 500 m moving mean and extreme values resulting from remaining artefacts from clouds or poorly co-registered strips in the annual mosaics are filtered out.

### 3.4 Estimating subglacial conditions

To assess potential spatial relationships between the locations where subglacial hydrologic pathways reach the grounding zone and align with ice-shelf channels, we derive a map of the subglacial hydraulic potential ($\phi$) based on observations (Fig. S6) as:

$$\Phi = \rho_w g z + \rho_i g H \qquad (3),$$

where $g$ is the acceleration due to gravity and $z$ and $H$ are equal to the BedMachine bed height and grounded ice thickness, respectively. Assuming that water is present everywhere at the bed, we again use the TopoToolbox FLOWobj function to compute the direction that water would flow along the gradient of the hydraulic potential, and the flow accumulation ("flowacc") function to find the cumulative number of pixels that contribute to flow in each downstream pixel, and convert this to the cumulative drainage area, or basal watershed area, for each pixel along the hydraulic potential gradient. While not intended as an actual estimate of subglacial discharge, this quantity provides a relative metric of where water is likely to be

routed in the subglacial system. We compare spatial patterns in inferred subglacial drainage at the grounding zone with the
occurrence of mapped ice-shelf channels.

**3.5 Uncertainty and sources of error**

Estimates of rates of changes in surface height, thickness, and basal mass are subject to uncertainties in the remotely-sensed
measurements, model outputs, and assumptions from which they are derived. Our methods follow those of Chartrand and
Howat (2020), which showed that uncertainties in $DH_E$/Dt and $M_b$ range from ~8–22 m yr$^{-1}$; this is similar to the variability in
our estimates (Table 1). We note that $M_s$ is derived from a temporal average of RACMO model output from only part of our
study period, which may omit the impact of anomalous precipitation events on our estimates of $M_b$. However, as we are
interested in the spatial variability of grounding zone change over several years, we do not expect the omission of short-term,
regional events to significantly impact our results as they will be partially captured in DSM surface heights.
Mapping of ice-shelf channel surface depressions and hydrostatic boundaries is subject to uncertainties arising from
the hydrostatic assumption, errors in manual delineation, and errors in the BedMachine bed height (Fig. S8). In particular, the
hydrostatic assumption may not be valid for portions of the TGIS (Chartrand & Howat, 2023), and an ice shelf's deviation
from hydrostatic balance may vary through time in the vicinity of ice-shelf channels (Chartrand & Howat, 2020; Stubblefield
et al., 2023). However, hydrostatic imbalance and temporal variations therein are estimated to be a fraction of ice thickness
(e.g. Chartrand & Howat, 2023; Stubblefield et al., 2023), and comparable to the BedMachine error in the vicinity and inland
of the IPY GL (Fig. S8; Morlighem et al., 2022). As we are interested in relative HB position through time, which occurs over
distances longer than ice thickness, rather than absolute HB position, we do not expect hydrostatic imbalance to impact our
interpretation of relative HB position inland of the IPY GL. However, the errors in BedMachine bed height increase rapidly to
~400 m between 2–5 km downstream of the IPY GL, so we are not as confident in absolute or relative HB position when it is
mapped downstream of the IPY GL. Furthermore, it should be noted that the IPY GL is derived from interferometric SAR
imagery and represents the inland limit of ice flexure, whereas the HBs represent the inland limit of hydrostatic balance, which
may differ from the limit of flexure by several km (e.g. Fricker et al., 2009). The IPY GL is therefore used only as a reference
from which to measure HB change, although it is fortuitous that the IPY GL (delineated from data primarily collected in 2007–
2009) represents the grounding line position near the beginning of DSM availability. As described in Section 3.1, manual input
is used to delineate the annual HB for each annual epoch, from which relative position is derived; the raw HB features are not
subject to manual error. Following masking to remove small, isolated HB features from consideration for the annual HB, points
along the most inland HB features from each year are manually selected so that the annual HB consistently represents the
grounding line in its most retreated position; we estimate that this manual delineation introduces independent errors < 200 m,
or 20 DSM strip pixels, in the position of any point along the annual HB.
Manual input for the ice-shelf channel surface depression positions is used to filter out spurious depressions in the
TopoToolbox output features, such as those that align with flow-perpendicular crevasses, and to define the reference channel
positions based on where a channel was consistently mapped by TopoToolbox in many DSMs. Thus, the reference channel
positions represent a manually-defined "average" of each ice-shelf channel's position through time and may not reflect its
position at any given time.

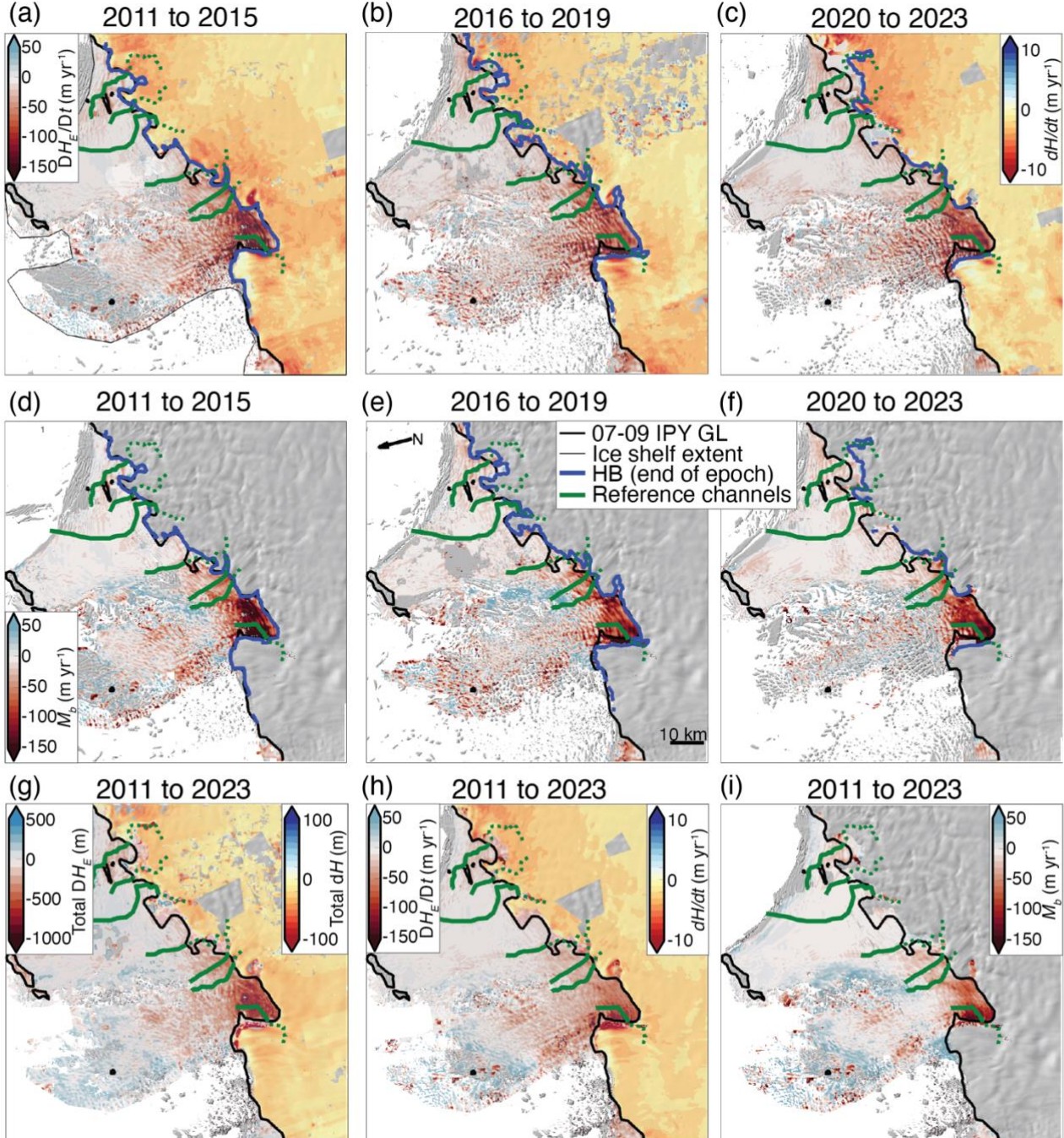

Figure 2: (a–c) Median Lagrangian rate of thickness change (D$H_E$/D$t$) on floating ice (dark red to blue colour scale) and Eulerian rate of thickness change (d$H$/d$t$) on grounded ice (dark orange to blue colour scale) for each 4–5 year epoch. (d–f) Median Lagrangian basal mass change rate ($M_b$, negative for basal melt) for each 4–5 year epoch. All maps (a–f) overlie the most recent annual mosaic hillshade from each epoch. (g) total Lagrangian thickness change (D$H_E$) on floating ice and total Eulerian thickness change (d$H$) on

**grounded ice for the entire study period. (h) Median D$H_E$/D$t$ and d$H$/d$t$ for the entire study period. (i) Median $M_b$ for the entire study**
**period. Maps for the entire study period (g–i) overlie the 200 m REMA mosaic. All maps show the IPY GL as a black curve and the**
**reference channels as green curves, and (a–f) show the most recent HB in each epoch as a blue curve.**

## 4 Results

The TGIS has REMA coverage from November 2010 to December 2022, enabling investigation of time-evolving HB and ice-
shelf channel positions as well as ice-column thinning and basal melt rates over the entire ice shelf at unprecedented spatial
resolution (Section 4.1). The HB retreated or was stagnant everywhere in the study area (Fig. 1) and ice-column thinning, basal
melting and grounded ice thinning dominated rates of change throughout the study period (Figs. 2, S7). We identify nine
regions of significant HB retreat and growth of basal cavities, labelled Cavities 1–9 (Fig. 1), discussed in more detail below
(Section 4.2). Seven ice-shelf channels that originate near the grounding zone are consistently identified throughout the study
period, labelled ThC1–7 (Fig. 1). Six of the channel locations align with inferred subglacial drainage routes (Figs. 3–5a), as
discussed in more detail below (Section 4.2).

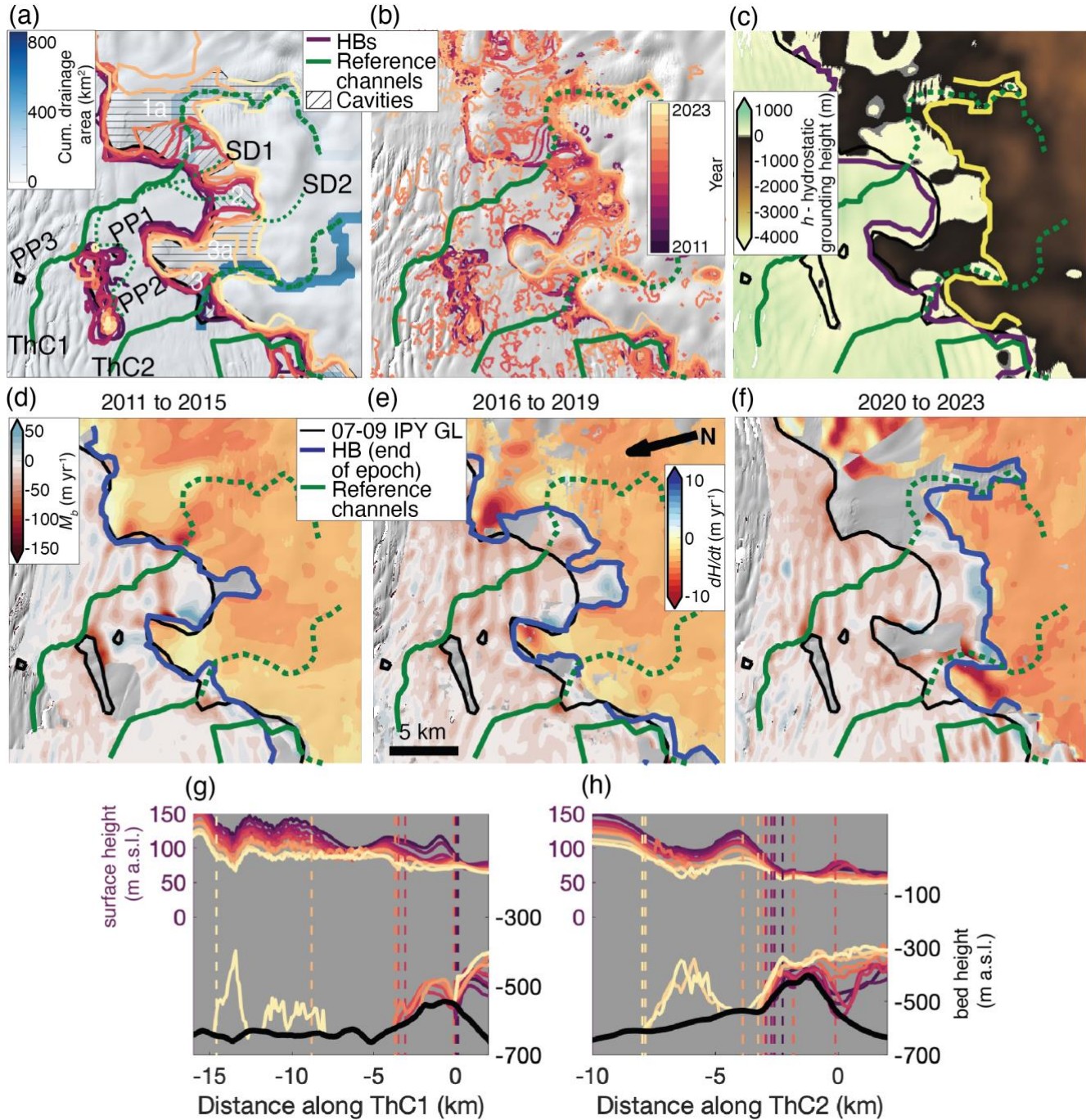


Figure 3: (a–f) Zoom on Box A from Fig. 1, showing reference channels ThC1–2 (thick green solid/dashed curves) and the 07–09 IPY GL (black curve). (a) Cumulative subglacial drainage area (blue colour scale, explained in Section 3.4) with the smoothed annual HBs (purple-orange-yellow curves, with less recent, darker features plotted below more recent features as lighter colours) and

Cavities 1–3 (hatched regions). Prominent surface depressions that are possibly connected to ThC1 are highlighted by thin dotted green curves. (b) Unfiltered HB features for each year (also on the purple-orange-yellow colour scale used for smoothed annual HBs). Panels (a) and (b) use the REMA v4 200 m mosaic hillshade as the base map. (c) Difference between the 2019 surface height and the hydrostatic grounding height (which is the flotation thickness plus the BedMachine v3 bed height) overlain by the smoothed annual HBs from 2011 (purple curve) and 2023 (yellow curve). (d–f) $M_b$ on the TGIS (dark red to blue colour scale) and $dH/dt$ on grounded ice (dark orange to blue colour scale) for each 4–5 year epoch overlain on the most recent annual REMA mosaic hillshade from each epoch. The most recent HB in each epoch is also plotted (blue curve). Annual surface height (left axis) and ice base (right axis) and BedMachine bed height (right axis, black curve) interpolated to reference channels (g) ThC1 and (h) ThC2. Vertical dashed lines mark the most landward intersection of each year's HB with the extended channel. Distances are defined from each channel's intersection with the IPY GL, with positive distance indicating advance and negative distance indicating retreat.

## 4.1 Time-evolving rates of basal mass change and ice thickness

Figures 2–5 indicate that rates of basal mass change are predominantly negative, indicating basal melting, but are spatially and temporally variable throughout the observation period, as are rates of floating ice-column and grounded ice thickness and surface-height change (Table 1). For ice shelf thickness changes in the Lagrangian frame ($DH_E/Dt$), thinning refers to change in the same column of ice as it advects with flow, rather than thinning at a fixed coordinate (Eulerian Frame), which we refer to on grounded ice. In general, ice-column thinning and basal melting accelerated from 2011–2015 to 2016–2019 and decelerated slightly in 2020–2023. The banded patterns of positive and negative values visible on the TEIS and TWIT in maps of $DH_E/Dt$ and $M_b$ may be due to changes in ice velocity not accounted for in flow-shifting using annual surface-velocity maps, or due to hydrostatic compensation around growing basal crevasses (e.g. Vaughan et al., 2012).

Overall, the TEIS experienced less basal melting than the TWIT. Nevertheless, there was some apparent mass gain in areas of TWIT, particularly in the downstream portion of the TWIT and the shear zone between the TEIS and TWIT (Figs. 2, 4–5). We expect that the apparent positive $M_b$ in the downstream portion of the TWIT may be an artefact of hydrostatic disequilibrium due to transient grounding, as evidenced by the presence of isolated HB features in that region (Figs. 5b, S4b). The fastest rates of ice-column thinning and basal melting on the TEIS consistently occur at the eastern ends of pinning points PP2 and PP4, with PP2 located near a zone of rapid HB retreat (Section 4.5) and the opening of Cavity 3 (Fig. 2, Sections 4.2.1 and 4.2.4, respectively).

The main trunk of the TWIT experienced the most intense basal melting at rates reaching 250 m yr$^{-1}$ in places near the grounding zone throughout the study period (Fig. 5). A closer look within Box B (Fig. 4d–f) shows that consistently high basal melt rates also occurred near the grounding zone in the vicinity of ThC5 and Cavity 7. There is also a flow-parallel band of accelerating ice-column thinning and basal melting along ThC6 near a zone of modest HB retreat along the most pronounced inferred subglacial drainage route (Figs. 2, 5d–f, Section 4.2.3).

Table 1. Rates of change in each annual mosaic epoch. All values are in units m yr$^{-1}$ (ice equivalent).

| | Rate of surface height change | | Rate of thickness change | | | | Rate of basal mass change | |
| | Floating $Dh/Dt$ | | Grounded $dH/dt$ | | Floating $DH_E/Dt$ | | Floating $M_b$ | |
| | Mean $\pm\ \sigma$ | median | Mean $\pm\ \sigma$ | median | Mean $\pm\ \sigma$ | median | Mean $\pm\ \sigma$ | median |
|---|---|---|---|---|---|---|---|---|
| Overall | $-0.7 \pm 4.9$ | $-0.9$ | $-2.2 \pm 0.7$ | $-2.1$ | $-14.2 \pm 27.7$ | $-8.6$ | $-6.2 \pm 27.9$ | $-3.1$ |
| 2011–2015 | $-1.0 \pm 6.3$ | $-0.8$ | $-2.8 \pm 1.6$ | $-2.6$ | $-21.4 \pm 36.9$ | $-11.0$ | $-14.5 \pm 37.4$ | $-6.2$ |
| 2016–2019 | $-3.0 \pm 5.0$ | $-2.0$ | $-2.4 \pm 2.2$ | $-2.3$ | $-27.3 \pm 35.6$ | $-17.8$ | $-19.0 \pm 37.6$ | $-11.1$ |
| 2020–2023 | $-2.7 \pm 5.1$ | $-1.8$ | $-2.0 \pm 2.0$ | $-1.9$ | $-26.2 \pm 35.3$ | $-16.2$ | $-16.8 \pm 34.8$ | $-9.2$ |


**4.2 Ice-shelf channel–HB interactions**
As mentioned above, the HB retreated or stagnated relative to its early positions everywhere by 2023, including on pinning
points, with significant variability in the rates of retreat, including some small and temporary areas of advance. In Cavities 6–
9, early HBs appear seaward of the 07–09 IPY GL, likely due to the differences in mapping method, but by 2023 the HB had
also retreated or stagnated relative to the IPY GL everywhere. While few consistent spatiotemporal patterns in HB retreat
emerged, there were no areas of sustained HB advance.
We identify several persistent basal incisions and surface depressions along seven ice-shelf channels. We reduce the
impact of noise in individual DSM strips on mapped basal incisions and surface depressions (Fig. S4) by defining fixed
reference locations of seven ice-shelf channels, called ThC1–7, to be where the greatest overlap of these features occurs. The
reference channels are extended above the grounding zone along surface depressions (where present) on grounded ice, enabling
us to measure changes in surface height (Figs. 2–5g–i), thickness (Fig. 6b), and velocity (Figs. 6 and 7) along the reference
channels, as well as changes in their intersections with the HB from each year relative to their intersections with the 07–09
IPY GL (Figs. 3–6), referred to as the ThCX–IPY GL intersection. IceBridge MCoRDS IPR ice-thickness profiles were queried
to verify the presence of the seven ice-shelf channels identified, although data quality along some profiles is poor (Figs. S4–
5). There are also several persistent surface depressions that extend inland of the IPY GL but do not appear aligned with
mapped ice-shelf channels, labelled SD1–7 (Figs. 3a, 5a).
A variety of behaviours and interactions between the ice-shelf channels and the grounding zone are observed. We
distil these into three major types:

1.  Narrow-cavity HB retreat along a narrow band parallel to an ice-shelf channel (ThC2, ThC3, ThC5; Section 4.2.1)
2.  Wide-cavity retreat along an ice-shelf channel (ThC1, ThC4; Section 4.2.2)
3.  Little to no HB retreat at the inland ends of ice-shelf channels (ThC6, ThC7; Section 4.2.3).

Figure 6a shows a time series of the change in position of the HB's intersection along each reference channel and illustrates
that Type 1 retreat tended to involve more steady retreat and Type 2 retreat tended to involve cycles of rapid retreat followed
by stabilisation. It is important to note that we do not consider these types to be mutually exclusive, as more than one type of
HB retreat is observed along several ice-shelf channels throughout the study period.
Several TGIS-wide similarities among ice-shelf channels are observed. The grounded ice within ~5 km of each
channel's inland end had background thinning rates between –1 to –4 m yr$^{-1}$ (Figs. 2a–c, 3–5d–f). All channels except for
ThC4 originate near areas of high cumulative subglacial drainage area at the grounding zone (Figs. 3–5a). Retreat of the HB
exceeding 1 km occurred along all reference channels except for ThC6 and ice-column thinning and basal melting occurred
near all channel intersections with the grounding zone within at least one multi-year epoch (Figs. 2–6). Notably, the mean
velocity along each HB is slightly slower at the ThCX–HB intersections than in non-channelized portions of the grounding
zone (Fig. 6c). However, no strong relationships emerge between changes in velocity and HB retreat rates along all channels;
notable correlations between changes in velocity and changes in HB position along individual channels are described in ensuing
sections (Fig. 7).

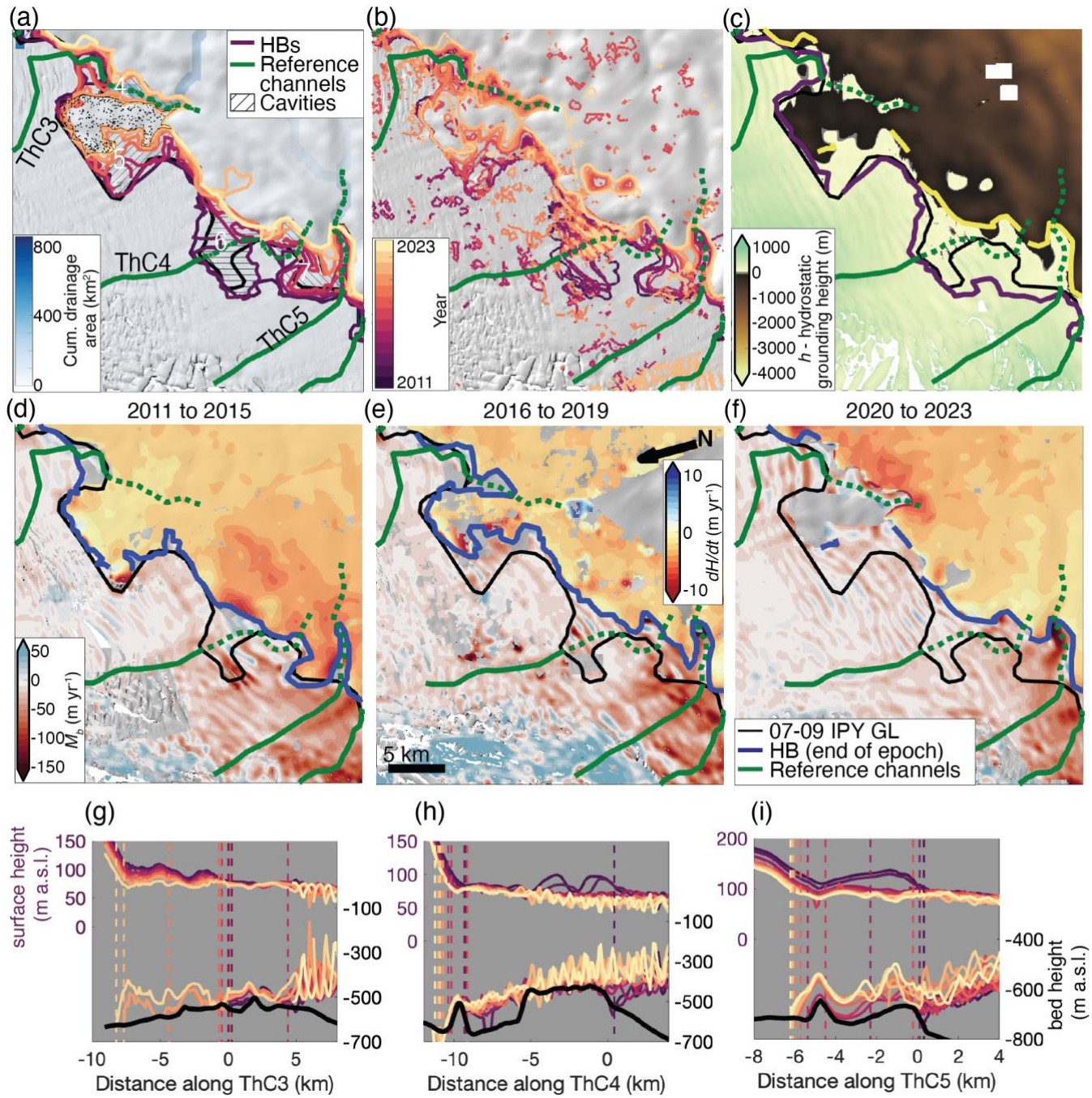


**Figure 4: (a–f)** Zoom on Box B from Fig. 1, showing reference channels ThC3–5 (thick green solid/dashed curves) and the 07–09 IPY GL (black curve). **(a)** Cumulative subglacial drainage area (blue colour scale) with the smoothed annual HBs (purple-orange-yellow curves, with less recent, darker features plotted below more recent features as lighter colours) and Cavities 4–7 (hatched regions). **(b)** Unfiltered HB features for each year (also on the purple-orange-yellow colour scale used for smoothed annual HBs). Panels (a) and (b) overlie the REMA 200 m mosaic hillshade. **(c)** Difference between the 2019 surface height and the hydrostatic grounding

**height overlain by the smoothed annual HBs from 2011 (purple curve) and 2023 (yellow curve). (d–f) $M_b$ on the TGIS (dark red to**
**blue colour scale) and d$H$/d$t$ on grounded ice (dark orange to blue colour scale) for each 4–5 year epoch overlain on the most recent**
**annual REMA mosaic hillshade from each epoch. The most recent HB in each epoch is also plotted (blue curve). Annual surface**
**height (left axis) and ice base (right axis) and BedMachine bed height (right axis, black curve) interpolated to reference channels (g)**
**ThC3, (h) ThC4, and (i) ThC5. Vertical dashed lines mark the most landward intersection of each year's HB with the extended**
**channel. Distances are defined from each channel's intersection with the IPY GL, with positive distance indicating advance and**
**negative distance indicating retreat.**

### 4.2.1 Type 1: Sustained retreat along narrow cavities

Channels ThC2, ThC3, and ThC5, as well as ThC1 in more recent years, were associated with steady, sustained HB retreat
along narrow cavities. These features are each directly aligned with an inferred subglacial drainage pathway (Figs. 3–5a) along
which HB retreat occurred, such that the cavities may strike oblique to the flow direction, but parallel to the surface depressions.
Furthermore, the profiles for these reference channels show large undulations in bed and surface height within 5–10 km of
their intersections with the IPY GL (Figs. 3–4g–i) which make it difficult to interpret D$H_E$/D$t$ and $M_b$ in these cavities because
the flow-shifting does not fully account for height changes due to horizontal advection, leading to alternating bands of apparent
ice-column thinning/basal melting and thickening/basal mass gain near the grounding zone (Figs. 3–4d–f).
The HB intersection with ThC2 predominantly exhibited Type 1 retreat as the HB retreated in a narrow band along
the sinuous ThC2 and its inferred underlying subglacial drainage route throughout the study period (Fig. 3). Cavity 3 widened
suddenly following a few years of retreat in a narrow band, temporarily exhibiting Type 2 characteristics, before growing
further inland in a narrow band. Between 2021–2023, continued HB retreat opened all of Cavity 3a so that it merged with
Cavity 2 (Fig. 3a), coinciding with a 20% increase in surface velocity along ThC2 after relatively steady speeds in earlier years
(Fig. 7). Figures 5c and 5h show that Cavity 3a overlies a bedrock ridge, and that the inland end of ThC2 overlies a deepening
trough. Rapid ice-column thinning and basal melt occurred near the intersection between ThC2 and the grounding zone
throughout the study period (Figs 2, 3d–f).
The HB position change along ThC3 exhibits the clearest example of Type 1 retreat (Figs. 1, 4). Figures 4a–b and 4g
show that the HB remained relatively stationary relative to ThC3 on a small bedrock ridge through 2018, before retreating
rapidly along ThC3 down a retrograde bed slope at a rate of 3.5 km yr$^{-1}$ between 2018–2020, opening the narrow Cavity 4
(Fig. 3a). HB retreat slowed in subsequent years despite the retrograde bed slope continuing inland. Despite the banded pattern
in the $M_b$ maps in this region, it appears that basal melt rates generally intensified throughout the study period (Figs. 4d–f). As
observed along ThC2, surface velocity along ThC3 accelerated between 2020–2023, although retreat had slowed by this time
(Fig. 7).
Figure 4 shows that the HB retreated south-eastward along ThC5 at an average rate of ~0.7 km yr$^{-1}$ between 2013–
2018, forming the finger-like southern portion of Cavity 7. During this time, ice-column thinning and basal melt rates reached
100 m yr$^{-1}$ and 60 m yr$^{-1}$, respectively (Figs. 4d–e), along ThC5 as the HB breached successive bedrock ridges before
stagnating on a relatively flat bed about 5–7 km inland of the ThC5–IPY GL intersection (Fig. 4i). Although the HB didn't
retreat much further along ThC5, Cavity 7 widened eastward and merged with Cavity 6. During this time, basal melt rates
within Cavity 7 intensified, exceeding 100 m yr$^{-1}$ along ThC5 (Fig. 4f).

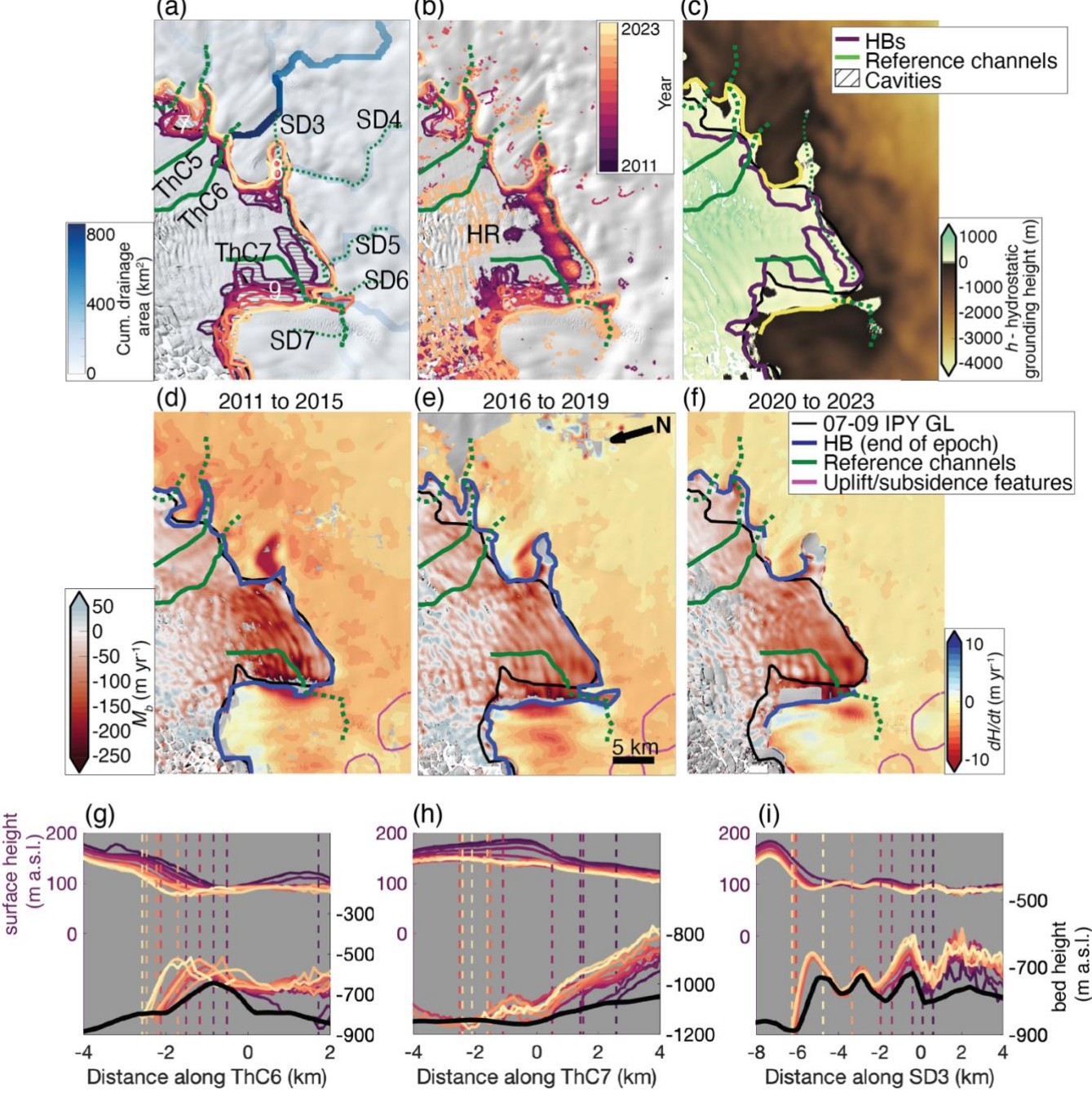

**Figure 5: (a–f)** Zoom on Box C from Fig. 1, showing reference channels ThC6–7 (thick green solid/dashed curves) and the 07–09
IPY GL (black curve). **(a)** Cumulative subglacial drainage area (blue colour scale) with the smoothed annual HBs (purple-orange-
yellow curves, with less recent, darker features plotted below more recent features as lighter colours) and Cavities 7–9 (hatched
regions). Prominent surface depressions SD3–7, some of which are possibly connected to ThC7, are highlighted by thin dotted green

curves. **(b)** Unfiltered HB features for each year (also on the purple-orange-yellow colour scale used for smoothed annual HBs), with an HB in the TWIT labelled "HR" for the "Holland Rumple" that was mapped by Holland et al., (2023). Panels (a) and (b) overlie the REMA 200 m mosaic hillshade. **(c)** Difference between the 2019 surface height and the hydrostatic grounding height overlain by the smoothed annual HBs from 2011 (purple curve) and 2023 (yellow curve). **(d–f)** $M_b$ on the TGIS (dark red to blue colour scale) and d$H$/d$t$ on grounded ice (dark orange to blue colour scale) for each 4–5 year epoch overlain on the most recent annual REMA mosaic hillshade from each epoch. The most recent HB in each epoch is also plotted (blue curve) and uplift/subsidence features identified by Rignot et al. (2024) (magenta curves). Annual surface height (left axis) and ice base (right axis) and BedMachine bed height (right axis, black curve) interpolated to reference channels **(g)** ThC6, **(h)** ThC6, and **(i)** SD3. Vertical dashed lines mark the most landward intersection of each year's HB with the extended channel. Distances are defined from each channel's intersection with the IPY GL, with positive distance indicating advance and negative distance indicating retreat.

### 4.2.2 Type 2: Wide-cavity retreat

Channels ThC1 and ThC4 are associated with sudden, rapid HB retreat off of bedrock highs to form wide Cavities 1a, 2, and 6, respectively (Figs. 3–4). Expansion of these relatively large cavities mostly occurred during 2013–2016 as widening across flow.

ThC1 and merging surface depressions SD1 and SD2 intersect the IPY GL in a region where the TEIS cavity is deeply embayed. Figure 3a shows that Cavity 1 opened up as the HB retreated suddenly along the ThC1 surface depression and an inferred subglacial channel between 2015 and 2016, and that Cavity 2 opened up as the HB retreated suddenly along SD2 (which does not align with an inferred subglacial channel) between 2014 and 2015, forming the lobes that make this region known as the "butterfly" region. At the same time, the velocity along ThC1 accelerated after a period of stability between 2011–2015 and continued to accelerate throughout the study period (Fig. 7). As basal melt rates accelerated to near 40 m yr$^{-1}$ by 2016–2019 (Fig. 3e), Cavities 1 and 2 widened but did not extend further inland (Fig. 3a). Cavity 2 merged with Cavity 1 between 2021–2022, eliminating the butterfly shape that was prominent in earlier years. After 2019, the HB exhibited Type 1 retreat, as the narrower Cavity 1a extended inland along the ThC1 surface depression and underlying inferred subglacial channel at a rate exceeding 2 km yr$^{-1}$ (Figs 3, 6a).

ThC4 is situated near the western edge of the TEIS and does not align with an inferred subglacial hydrological route. ThC4 appears to result from the merging of two incisions initiated at two bedrock ridges, the wider of which is located at –5 to 0 km along ThC4 in Figs. 4h and 7 (where 0 km is the northernmost ThC4–IPY GL intersection); the narrower ridge is located further south at about –9.2 km in Figs. 4h and 7, near the western end of Cavity 7 (Fig. 4a). Several instances of the sinuous surface depression persist across Cavities 6 and 7 (Fig. S4c), while instances of the basal incision appear to curve around Cavity 6 (Fig. S4a). Figure 4h shows that Cavity 6 opened along ThC4 between 2011–2015 as ice ungrounded from the wide ridge. In subsequent years, Cavity 7 reached its eastern and southern maximum extents and merged with Cavity 6 as the HB retreated off the narrower southern bedrock ridge (Figs. 4a, 4h). Despite basal melt rates consistently exceeding 60 m yr$^{-1}$ near ThC4 (Fig 4d–f, which may be unreliable due to possible intermittent re-grounding, indicated by the isolated HBs in Fig. 4b), the HB did not retreat much further along ThC4, stabilising within a bedrock trough (Fig. 4h). The velocity along ThC4 was highly variable, decelerating by about 5% as Cavity 6 grew, accelerating by about 10% between 2015–2016, slowing

again between 2016–2019 as Cavity 7 grew, and speeding up, especially downstream of Cavity 6, as the HB stagnated (Fig.
7).


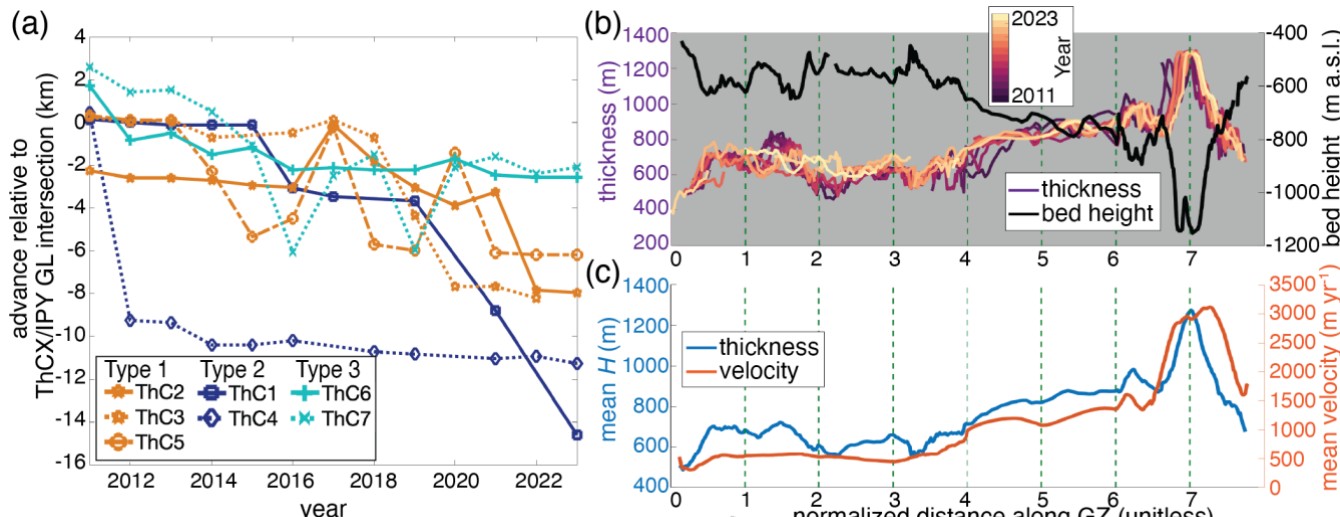


**Figure 6: (a) Time series of HB position along each reference channel, with negative distance indicating retreat, and 0 km marking**
**the ThCX/IPY GL intersection. The orange time series indicate Type 1 retreat, the blue time series indicate Type 2 retreat, and the**
**teal time series indicate Type 3 retreat. (b) Thickness along each annual HB and BedMachine bed height along the IPY GL and (c)**
**mean thickness and velocity along the IPY GL, with unitless distance normalised to the intersection of each year's HB or the IPY**
**GL with each ice-shelf channel ThC1–7.**

### 4.2.3 Type 3: Little to no HB retreat

Ice-shelf channels ThC6 and ThC7 were associated with modest HB retreat that did not fit into Type 1 or Type 2 cavity shapes,
despite their alignment with inferred subglacial drainage routes (Fig. 5).
ThC6 is aligned with the strongest inferred subglacial channel just east of the TWIT main trunk, but the surface
depression does not appear to extend inland of the IPY GL (Figs. 4a, S4c). Thus, we manually extended the landward end of
the ThC6 reference channel about 5 km inland of the grounding zone to show retreat past the IPY GL. Throughout the study
period, rapid basal melt and grounded and floating ice thinning occurred near the ThC6–IPY GL intersection (Fig. 5d–g). The
HB retreated at a rate of about 0.3 km yr$^{-1}$ along ThC6 (Fig. 6a), and the small cavity forms a v-shape along the inferred
subglacial channel (Fig. 5a), potentially indicating that Type 1 retreat will occur in the future. ThC6 also experienced among
the widest fluctuations in velocity, although there was no clear relationship between velocity and HB retreat (Fig. 7).
The western flank of the TWIT main trunk grounding zone exhibits complex morphology and changes, but relatively
slow rates of retreat in the vicinity of ThC7 and merging surface depressions SD6–7 (Fig. 5a). At ThC7, the HB retreated ~0.4
km yr$^{-1}$ as Cavity 9 extended westward between 2013–2023 (Fig. 6a), with basal melt rates consistently exceeding 100 m yr$^{-}$
[1] (Fig. 5). Notably, in 2016 and 2019, the HB temporarily retreated along a narrow band along the ThC7 surface depression
(Fig. 5i), within the same timeframe as a 300 m yr$^{-1}$, or 11%, increase in velocity between 2016–2020 (Fig. 7).

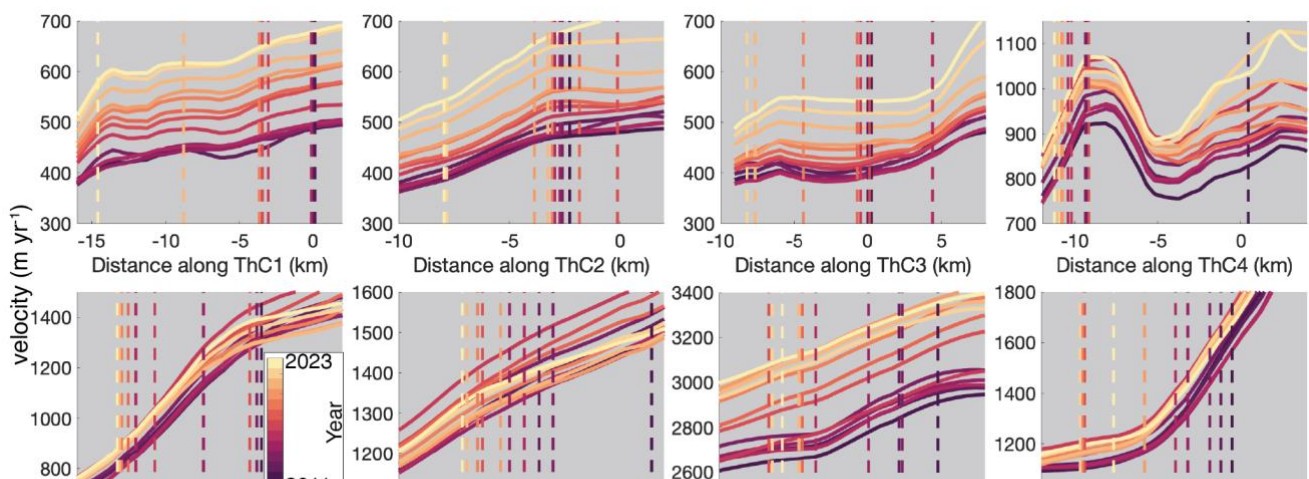


**Figure 7: MEaSUREs annual velocity (2011–2015) and averaged summer quarterly velocities from InSAR (2016–2023) interpolated to reference channels ThC1–7 and SD3. Vertical dashed lines mark the most landward intersection of each year's HB with the reference channel. Distances are defined from each channel's intersection with the IPY GL, with positive distance indicating advance and negative distance indicating retreat.**


### 4.2.3 Retreat not associated with ice-shelf channels

There are a few regions where HB retreat is observed in the absence of ice-shelf channels and/or inferred subglacial drainage
routes. Between ThC3 and ThC4, there is a region where the HB shifts eastward between 2013 and 2021 at a rate of up to 0.6
km yr$^{-1}$, opening Cavity 5 (Fig. 4). Notably, the 2022 HB connects with Cavity 4, indicating that a larger cavity may have
opened, but there was insufficient coverage to map the 2023 HB in this region (stippled area in Fig. 4a). In 2022, this region
only has coverage from one or two strips (Fig. S2), resulting in only two mappings of HB features from which the annual HB
was manually delineated. Although some regions are covered by few strips in several years, we are more confident in HB
positions that persist or display a pattern over several years, and one year of data does not provide sufficient evidence to
conclude that this entire region was ungrounded in 2022.

The main trunk of the TWIT exhibited complex HB changes seemingly independent of ThC7. Merging surface

depressions SD3 and SD4 are identified inland of the south-eastern corner of the embayed grounding zone in the main trunk
and appear to be associated with HB retreat (Fig. 5a, S4c). SD4 aligns with an inferred subglacial channel, but SD3 does not

(Fig. 5a). We also identify a surface depression extending from SD3 and SD4 parallel to the IPY GL at the southern grounding zone of the main trunk, but IPR transects M6 and M7 do not indicate that there are corresponding basal incisions (Fig. S5). The surface depression extending across the main trunk may instead be a dynamical response to the transition of flow off the ridge along the southern grounding zone of the main trunk (Fig. S8a). Figures 5 and 6a show that the HB retreated along SD3 at an average rate of ~0.8 km yr$^{-1}$ between 2011–2018, opening the narrow Cavity 8 along an undulating bedrock topography. The fastest retreat rates (~2 km yr$^{-1}$) occurred between 2015–2017, followed by relative stability over a bathymetric low after 2018. Ice-column thinning and basal melt rates were consistently high at the downstream end of Cavity 8, exceeding 100 m yr$^{-1}$ along the eastern flank of the main trunk in all three multiyear epochs (Figs. 2, 5d–f).

Figure 5b shows that the TWIT main trunk contained many small, isolated HBs throughout most of the study period that may indicate intermittent grounding, although the bed height is unreliable here due to the use of indirect measurements for bed heights in ice-shelf cavities (Fig. S8, Morlighem et al., 2020). The annual HBs from the early years of the study period extended eastward in a narrow band between ThC7 and the IPY GL, narrowing the ice-shelf cavity in the centre of the main trunk to only 2.5 km (Figs. 5a–b). By 2013, Cavity 9 had opened and the HB was approximately at the same location as the IPY GL along the southern grounding zone of the main trunk. Furthermore, the HB at the western flank retreated steadily to the west and up a ridge in the basal topography throughout the study period (Figs. 5a, S8a). The western flank of the main trunk also experienced high rates of ice-column thinning and basal melting throughout the study period as Cavity 9 opened (Figs. 2, 5d–f).

## 4.3 Pinning points

The HBs at the pinning points exhibited a variety of behaviours. Our HBs maps did not capture PP3, although PP1, PP2, and PP4–6 were mapped. The IPY GL did not contain a pinning point in the main trunk of the TWIT, although Holland et al. (2023) track the evolution of an ice rumple near the centre of the main trunk, which disappeared between 2011–2022. We only map PP1 through 2014, and PP2 grew smaller through 2023 (Fig. 3). Pinning points 1 and 2 experienced relatively high rates of ice-column thinning and basal melting at the eastern extent of the IPY GL of PP2 (Figs. 2, 3d–f). Furthermore, ThC1 possibly rerouted as these pinning points shrank; from 2011–2014, the surface and basal manifestations of ThC1 curved toward the west, following the western prong of the "y" shape south of PP1, then straightened toward the eastern prong of the "y" shape between 2015–2022 (Fig S4).

Pinning points 4 and 5 were mapped throughout the study period, without much change in HB position, and the surrounding ice shelf experienced ice-column thinning and basal melting at rates similar to the rest of the TEIS (Figs. 1–2). We observe possible north-westward growth of PP4 and PP5 but note that BedMachine is poorly constrained here (Fig. S8). Pinning point 6 was also mapped throughout the study period, although it is largely indistinguishable from other small, noisy HBs that are mapped, but filtered out, on the bedrock high in the TWIT (Figs. 1, S4d).

As discussed in Section 4.2.3, the unfiltered HBs in the TWIT (Fig. 5b) indicate the presence of many small pinning

points, which appear to shrink or disappear over time as the ice thins. We also map an isolated HB near the ice rumple mapped
by Holland et al. (2023; labelled "HR" for "Holland Rumple" in Fig. 5b) which disappears by 2014. However, as noted
elsewhere, the bed topography is poorly constrained in this cavity so the locations of HBs and basal incisions inferred using
the hydrostatic assumption are uncertain. Notably, the TWIT lost an area of ~1270 km$^2$ between 2011–2012, retreating from
potential pinning points near the front, and continued to lose area throughout the study period (Fig. S1).

## 5 Discussion

This work reveals high-resolution observations of important processes affecting the shape and structure of the Thwaites Glacier
and TGIS. We observe evidence for high rates of grounding zone retreat along ice-shelf channels and inland subglacial
channels on the Thwaites Glacier and TGIS using REMA DSMs to map ice-shelf channels, surface depressions, and rates of
thickness and basal mass change. We observe three major types of retreat along seven ice-shelf channels and associated surface
depressions: narrow-cavity retreat, wide-cavity retreat, and little to no retreat. Regions associated with each type of retreat are
often collocated with high rates of ice-shelf basal mass loss and ice-column thinning and down-stream of the grounding zone
and grounded ice thinning inland, particularly in the 2011–2015 epoch (Fig. 2).

### 5.1 Basal melt rates

Aside from some differences in magnitude, the general patterns of persistent HB retreat and rapid ice-column thinning and
basal melt along the TGIS grounding zone that we observe are in agreement with other recent observations (e.g. Holland et
al., 2023; Milillo et al., 2019; Schmidt et al., 2023; Adusumilli et al., 2020). All confirm that $M_b$ is consistently smaller in
magnitude on the TEIS than the TWIT, and that more basal melting occurs near the grounding zone than further seaward.
Others have also observed and modelled rapid and potentially unstable retreat of the grounding zone, attributed to enhanced
basal melting (Joughin et al., 2014; Rignot et al., 2014; Seroussi et al., 2017; Yu et al., 2018; Milillo et al., 2019; Hoffman et
al., 2019). Enhanced basal melt rates are in turn attributed to the intrusion of warm Circumpolar Deep Water (CDW) flowing
along bathymetric troughs to the grounding zone (Nakayama et al., 2018; Milillo et al., 2019; Hogan et al., 2020). In the TGIS
region, CDW intrusion primarily occurs along two bathymetric troughs (indicated in Fig. 1), allowing it to reach the grounding
zone of both the TEIS and the TWIT (Dutrieux et al., 2014; Dotto et al., 2022).

The modest basal melt rates that we observe in the vicinity of Cavities 1, 1a, and 2 (Fig. 3d–f) are largely in agreement

with those observed by the Icefin submersible in the same region (Schmidt et al., 2023). Holland et al. (2023) show modest
apparent basal mass gain along the eastern and southern flanks of the TWIT main trunk in 2011 and basal melt rates reaching
250 m yr$^{-1}$ along the southwestern boundary in both 2011 and 2022. High rates of apparent basal mass gain in the TWIT main
trunk are also inferred by Milillo et al. (2019). We observe basal melt rates reaching 150 m yr$^{-1}$ throughout the main trunk,
especially at the western flank, but we observe no basal mass gain at the southern flank. We suggest that the choice of
Lagrangian flow-shifting methods can result in apparent mass gain in the TWIT if the time-evolving flow divergence is not
accounted for (Fig. S9). Milillo et al. (2019) posit that, as the ice thins and the grounding line retreats, the bending zone where
the ice is deflected below flotation before rebounding also retreats, causing changes at the surface to mask the true magnitude
of ice thinning and overestimate $M_b$. With the caveat that BedMachine is poorly constrained in this ice-shelf cavity (Fig. S8)
we also see intermittent re-grounding of ice in the raw HB features (Figs. 5b, S4d), which would further complicate the actual
hydrostatic rebound, as well as the hydrostatic assumption and assumptions about ice flow. These factors all reduce confidence
in the inferred $DH_E/Dt$ and $M_b$ in the TWIT main trunk. While we do not estimate melt rates below grounded ice, we observe
a few regions where isolated HBs inland of the continental HB are aligned or collocated with inferred subglacial channels and
regions of grounded ice thinning (e.g. in and around Cavities 1a, 6, 8, and 9); these resemble regions of uplift and subsidence
mapped by Rignot et al. (2024) which may indicate enhanced subglacial melting upstream of the grounding zone and are
discussed further in Section 5.2.
**5.2 Hydrostatic boundaries**
In agreement with other studies, we find a mix of stagnation and retreat of the HB along the entire coast of the TGIS. The
fastest retreat rates are collocated with retrograde slopes in the bed topography, ice-shelf channels that intersect the IPY GL
and/or the positions of inferred subglacial channels, and high basal melt rates. Notably, our $M_b$ estimates and HB retreat for
the fast-flowing TWIT main trunk align closely with several other studies; we find HB retreat rates of 0.3–0.6 km yr$^{-1}$ between
2011–2019 and basal melt rates reaching 180 m yr$^{-1}$ as Cavity 9 opened along the western flank.  Milillo et al. (2019) showed
that the grounding line along the western flank retreated at a rate of 0.6 km yr$^{-1}$ to the west between 2011 and 2017. Our results
also align with those of Bevan et al. (2021), who documented the opening of an ice-shelf cavity along the retreating western
flank between 2014 and 2017, and Rignot et al. (2024), who observe a retreat rate of about 0.5 km yr$^{-1}$ between 2018–2023 in
this region. Milillo et al. (2019) attributed the rapid ungrounding at the point labelled "A" in their figures (which falls within
Cavity 9) to its prograde slope, which favours CDW intrusion and efficient cavity opening, consistent with plume theory
(Jenkins, 2011).
ThC2, ThC3, ThC5, and SD3 are associated with Type 1 HB retreat in narrow bands oblique to the flow direction,
but parallel to inferred subglacial channels, lending confidence to our predicted subglacial channel distribution and indicating
that subglacial melting is strong (Figs. 3–5). Similar retreat along subglacial channels has been observed on
Nioghalvfjerdsfjorden Glacier (N79) Ice Tongue in northeast Greenland (Narkevic et al., 2023) and the Petermann Glacier Ice
Tongue in northwest Greenland (Ciracì et al., 2023); in both cases, retreat occurred in narrow bands aligned with the direction
of ice flow. Hager et al. (2022) showed that the inclusion of channelized drainage into their model increased effective pressures
in non-channelized regions near the grounding line, which may increase basal drag and reduce grounding line retreat and mass
loss (Yu et al., 2018) and velocities (Gillet–Chaulet et al., 2016; Joughin et al., 2019). We observe little to no retreat where
subglacial channelization is not present, which may be due to high points or prograde slopes in the bed topography but could

possibly be due in part to enhanced basal friction in the absence of subglacial water or its concentration within subglacial channels. It is expected that subglacial melt rates are higher where discharge of subglacial meltwater occurs (e.g. Le Brocq et al., 2013; Washam et al., 2019). The basal meltwater volume has been estimated at 3.5 Gt yr$^{-1}$ for the 189,000 km$^2$ Thwaites Glacier drainage basin, with most of the melt occurring within about 50 km of the grounding zone (Joughin et al., 2009). Our study area extends from ~10–100 km inland of the grounding zone, so ample subglacial water is available, and may discharge in the manner we predict (Figs. 3–5a, S4b), forming a collection of ice-shelf channels when it reaches the ice shelf (Section 5.3). While we do not investigate evolution of the subglacial cumulative drainage area over time, we posit that any discrepancies in orientation or position among mapped surface depressions and basal incisions may be due to rerouting of the subglacial drainage system.

In contrast with the retreat observed along the continuous continental grounding zone and shrinking or ungrounding of pinning points 1, 2, and 3, PP4 and PP5 exhibit signs of growth throughout the study period, particularly with advance to the northwest of their IPY positions (Figs. 1, S4) Indeed, the bed topographic high on which these pinning points rest extends and grows taller to the northwest (Fig. S8a), and some localised thickening is observed as the TEIS flows onto PP5, although the region is dominated by ice-column thinning and basal melting (Fig. 2). Due to gaps in coverage (Fig. S2), it is difficult to tell whether the ice-shelf area to the north of the pinning points is changing. Wild et al. (2022) demonstrate that although PP5 is structurally sounder than PP4, the two used to be connected and their separation and the disconnection of the TEIS and TWIT has altered ice flow. This change, along with the advection of thinner and more damaged ice on the TEIS portends ungrounding from the pinning point within the next decade (Wild et al., 2022).

The unfiltered and unsmoothed HBs observed throughout the Thwaites Glacier and TGIS provide insight into potential future behaviour. We observe isolated HBs inland of the continental grounding zone, indicating that the ice surface is below the "hydrostatic grounding height" (ice surface height resulting from adding the flotation thickness $H_E$ for all ice to the bed height, Figs. 3–5b–c) above bedrock lows. The bed heights from BedMachine v3 are relatively reliable inland of the IPY GL, with errors < 50 m (Fig. S8b), which is similar to the uncertainty in our calculation of $H_E$, promoting confidence in the existence of pockets where the surface is below the hydrostatic grounding height inland of the grounding zone. Several of the isolated HBs inland of the grounding zone persist over multiple epochs and align with inferred subglacial drainage pathways. The isolated HBs inland of the IPY GL are necessarily located at bed topographic lows, and likely contribute to the cycle of rapid retreat and temporary stabilisation observed along several reference channels. Indeed, Figs. 3–5b show that the continental HB retreated far enough for several cavities to encompass some of these isolated HBs from earlier epochs. Notably, the grounding zone near ThC6, which aligns with the inferred subglacial channel location with the highest flow accumulation, experienced modest HB retreat but high rates of grounded ice thinning and ice-shelf basal melting and ice-column thinning (Fig. 5), potentially foreshadowing the formation of a Type 1 cavity; however, there are few isolated HBs further inland (Fig. 5b–c).

We also observe isolated HBs throughout the TWIT, indicating that the ice surface is above the hydrostatic grounding height above bedrock highs (Figs. 3–5b–c). In contrast to our certainty in mapping isolated HBs above the grounding zone,

BedMachine errors increase rapidly to 400 m downstream of the IPY GL, where the bed is inferred from gravity inversion, so
we are less confident in the existence of additional pinning points within the TGIS, with the exception of the "HR" ice rumple
which was independently mapped by Holland et al. (2023) using similar methods (Fig. 5b). We expect that the isolated HBs
that persist on high points within ~2 km downstream of the IPY GL throughout several years (where BedMachine errors are
around 100 m, Fig. S8b), may have been or currently are pinning points. Likewise, we expect sufficiently high points between
the isolated HBs inland of the grounding zone (Figs. 3–5b, S8) to serve as temporary pinning points as new cavities open
around them as the continental HB retreats.
**5.3 Ice-shelf channels**
Based on our inferred subglacial drainage pattern and mapped surface depressions and basal incisions, we suggest that all ice-
shelf channels identified in this study except ThC4 are subglacially sourced. Ice-shelf channels have been mapped previously
on the TGIS by Alley et al. (2016), and several subglacial channels were also identified by Milillo et al. (2019), many of which
align with our DSM-derived channel positions. Comparisons between these observations provide insights into the formation
of each ice-shelf channel.

There is relatively strong evidence that ThC1 is a subglacially-sourced ice-shelf channel. The downstream end of

ThC1 is about 6 km away from an ice-shelf channel identified by Alley et al. (2016) (Fig. S10), and its inland end roughly
aligns with where Milillo et al. (2019) document the formation of an approximately 1 km wide subglacial channel near Cavities
1, 1a, and 2 (points C and D from Fig. 1 in Milillo et al. (2019)) before the grounding line retreated to its 2017 extent. They
observed no change in velocity along the subglacial channel, and thus attribute thinning in this region to ocean-induced basal
melting rather than dynamic thinning (Milillo et al., 2019; Millgate et al., 2013). Schmidt et al. (2023) confirmed strong basal
melting in this region, with the fastest rates along the steep slopes of terraces at the ice-shelf base, consistent with observations
at Pine Island Glacier (Dutrieux et al., 2014). Although Schmidt et al. (2023) did not sample at the location of ThC1, their
finding that the greatest basal melting occurs along steep basal slopes in this region provides further evidence that ThC1 is a
subglacially-sourced channel whose steep sides promote high basal melt rates and retreat along its trunk.

Two channels mapped by Alley et al. (2016) roughly align with the locations of ThC3 and ThC4, and a third runs

parallel to the end of ThC7 but begins further downstream (Fig. S10). Alley et al. (2016) considered the ice-shelf channels
parallel to ThC4 and ThC7 to be subglacially sourced. Our observations support this claim for ThC7 but suggest that ThC4
may be a grounding-zone sourced incision as ice flows over local bedrock topographic highs as described in Sections 4.2.2
(Fig. 5), although we cannot confirm whether it entrains buoyant plumes. Furthermore, the retreat along SD3 also appears to
be coincident with a subglacial drainage channel modelled and mapped by Hager et al. (2022, Fig. 5), although due to the
breakdown of the hydrostatic assumption in the TWIT main trunk, we cannot confirm whether this inferred subglacial channel
forms an ice-shelf channel in the ice shelf.

One of the channels we observe, ThC7, initiates near where two subglacial drainage channels discharge to the ocean

(Rignot et al., 2024). Using differential SAR interferometry, Rignot et al. (2024) observed several circular areas ~4–6 km in
diameter with time-varying uplift and subsidence (10–20 cm). These features are located above subglacial topographic
depressions that abut km-scale subglacial ridges. The major features are all adjacent to prominent subglacial drainage channels
and resemble the isolated HBs we infer inland of the grounding zone in and around Cavities 1a, 6, 8, and 9 (Figs. 3–5b). Rignot
et al. (2024) conclude that the filling and draining of the more inland features is driven by fluctuations in the subglacial water
flow through the nearby channels. For the large 'bull's eye' feature just ~6 km above the grounding zone (see Fig. 4c in Rignot
et al. 2024), however, they speculate that the vertical motion is due to tidally-forced seawater intrusion, which they suggest
should cause enhanced subglacial melting. They do not specify the magnitude of this melt other than to say it should be much
lower than 20 m yr$^{-1}$. If this non-steady melting is significantly above the background subglacial melt rate, we would expect
to see a signature in the long-term thinning rates. Instead, the 2020–2023 elevation change data show thinning of 1–2 m yr$^{-1}$
in the area surrounding the feature near ThC7 with minor thickening ($< 0.5$ m yr$^{-1}$) near its centre in 2020–2023, providing
little or no indication of enhanced subglacial melt (Figs. 5, S7). We also note that d$H$/d$t$ derived from annual DSM mosaics
does not provide the fine temporal resolution (up to sub-daily) over which uplift/subsidence features were observed in this
study. We do not observe increased rates of thinning for most of these closed regions, even when they are near the main HB,
suggesting that any enhanced subglacial melting due to incursion of seawater may not persist long enough to significantly
impact the signal on multi-annual timescales for most of the glacier. Furthermore, Bradley and Hewitt (2024) show through
modelling that Thwaites Glacier is likely not susceptible to runaway melting as a result of seawater intrusion processes. An
alternate hypothesis is that all of the circular features are driven by subglacial water flow rather than seawater intrusion. This
hypothesis is supported by a strong gradient in the hydraulic potential between the grounding zone and the 'bull's eye' feature,
which should drive the water toward – not away from – the ocean (Fig. S6). Seawater intrusion is also problematic because it
needs to occur over an area where the predominant flow direction should be seaward to accommodate major subglacial
outflows. These features likely fill and drain through exchange of water with the adjacent subglacial channels, similar to how
lakes located much farther inland fill and drain below Thwaites Glacier (Smith et al., 2017) and Jutulstraumen Glacier (Neckel
et al., 2021). If this is the case, the pressure boundary condition where these channels meet the ocean should be subject to tidal
modulation (10 kPa) sufficient to explain the observed ~10–20 cm uplift/subsidence (1–2 kPa).

## 6 Conclusions

This study presents novel, time-evolving rates of ice-shelf thickness and basal mass change and proxies for grounding line and
ice-shelf channel position on the TGIS derived from high resolution REMA DSM products, providing further evidence linking
high basal melt rates along ice-shelf channels to rapid rates of grounding-zone retreat (e.g. Narkevic et al., 2023; Holland et
al., 2023; Ciracì et al., 2023). Hydrostatic boundary retreat rates averaging 0.6 km yr$^{-1}$ and at times $> 3$ km yr$^{-1}$ were observed
concurrently with persistent ice-shelf channels and basal melt rates as high as 250 m yr$^{-1}$. The retreat is not fully attributable
to the presence of ice-shelf channels, as several regions where HB retreated along reference channels also had deep retrograde
bed slopes and/or were likely to be in contact with warm CDW. This study does not deconvolve all potential causes and effects
of HB retreat, such as changes in ice velocity through time (e.g. dos Santos et al., 2021), varying subglacial discharge (e.g.
Hager et al., 2022), or changing ocean currents (e.g. Holland et al., 2023; Dotto et al., 2022), but supports the hypothesis that
ice-shelf channels, whether initiated at the grounding zone or subglacially, are associated with more rapid grounding zone
retreat than non-channelized areas. Our observations are consistent with other work that suggests buoyant meltwater plumes
can entrain CDW to form plumes with strong basal melting capabilities (e.g. Le Brocq et al., 2013).

These results also provide additional evidence for the recent opening of new ice-shelf cavities not associated with ice-

shelf channels, as observed by Milillo et al. (2019), Bevan et al. (2021), and Schmidt et al. (2023), and point to the potential
for continued, unstable retreat of the grounding zone (e.g. Yu et al., 2018; Joughin et al., 2014), particularly along inferred
subglacial drainage pathways. As the grounding zone continues to retreat and subglacial pressures change, we suggest that
retreat along existing and/or rerouted subglacial channels that intersect the grounding zone will continue to form narrow, Type
1 cavities in the future, complicating the task of accurately predicting future grounding zone retreat.

Milillo et al. (2019) point out that several of the newly opened ice-shelf cavities have less than 100 m between the

ice-shelf base and the sea floor, and to simulate these basal melt and retreat processes would require a significantly finer spatial
resolution than is currently available to ocean models. This methodology can be applied to other ice shelves to further
investigate the prevalence of HB retreat at channelized and non-channelized grounding zones to further investigate relevant
changes in ice-shelf structure, velocity, basal and subglacial melt rates, and subglacial drainage. This study is an important
step towards better understanding these complex and critical regions of the Antarctic ice sheet and the relevant temporal and
spatial scales over which these processes occur.
**Data and Code Availability**
REMA v4.1 2 m strips (DOI:10.7910/DVN/X7NDNY) and 200 m mosaics (DOI: 10.7910/DVN/EBW8UC) are available at
the Polar Geospatial Center. The following datasets are available at NSIDC DAAC: BedMachine Antarctica V003 bed heights,
firn and ice thicknesses, Eigen–6C4 geoid data (DOI: 10.5067/FPSU0V1MWUB6), MEaSUREs Antarctic Boundaries for IPY
2007–2009 from Satellite Radar V002 (DOI: 10.5067/AXE4121732AD), MEaSUREs Antarctic Grounding Line from
Differential Satellite Radar Interferometry V002 (DOI: 10.5067/IKBWW4RYHF1Q), MEaSUREs InSAR-Based Antarctica
Ice Velocity Map V002 (DOI: 10.5067/D7GK8F5J8M8R), MEaSUREs Annual Antarctic Ice Velocity Maps V001 (DOI:
10.5067/9T4EPQXTJYW9), ATLAS/ICESat–2 L3A Land Ice Height V005 (DOI: 10.5067/ATLAS/ATL06.005) and V006
(DOI: 10.5067/ATLAS/ATL06.006), IceBridge MCoRDS L2 Ice Thickness V001 (DOI: 10.5067/GDQ0CUCVTE2Q),
IceBridge ATM L1B Elevation and Return Strength V002 (DOI: 10.5067/19SIM5TXKPGT), IceBridge LVIS–GH L2
Geolocated Surface Elevation Product V001 (DOI: 10.5067/RELPCEXB0MY3), and IceBridge Riegl Laser Altimeter L2
Geolocated Surface Elevation Triplets V001 (DOI: 10.5067/JV9DENETK13E). The DTU22 MDT model is available at

(2019). The CATS2008b tide model (DOI: 10.15784/601235) is available at USAP–DC. RACMO 3p2 data are available at https://www.projects.science.uu.nl/iceclimate/publications/data/2018/vwessem2018_tc/RACMO_Yearly/. TopoToolbox v2.3.1 is available on the Mathworks File Exchange.

All code, gridded products generated in this study (annual mosaics, annual velocities derived from the Amundsen Sea quarterly velocities, and rates of change), and shapefiles of the reference channels, HBs, and strips with registration information are freely available at DOI: 10.5281/zenodo.13667120.

## Author Contributions

AC conceived the ideas and carried out analyses with support from IH. IH created the annual REMA DSM mosaics, IJ provided the quarterly velocity maps for the Amundsen Sea regions, and BS performed the CryoSat–2 registrations for the REMA DSM strips. AC prepared the manuscript with contributions from all authors.

## Competing Interests

Some authors are members of the editorial board of The Cryosphere.

## Acknowledgements

This work was funded by the National Aeronautics and Space Administration (NASA) Future Investigators of NASA Earth and Space Science and Technology grant no. 80NSSC20K1658. Allison Chartrand and Ian Howat were also supported by the National Science Foundation (NSF) award no. 2217574, and the Ohio State University. Ian Joughin was funded by NASA grant no. 80NSSC20K0954, and Ben Smith was supported by NASA grants no, 80NSSC20K1064 and no. 80NSSC22K1107.

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
