# Peer review of "intersect its grounding zone"

_EGUsphere, 2024_

## Author Comment (AC1)

**Response to Reviewer 1 (Adrian Luckman)**

**Initial comments**

This study uses a time-series of high resolution DSMs to investigate the interplay between grounded ice, ice shelf, floating tongue and basal channels at Thwaites Glacier. This is a topic of great interest in the cryosphere and the research is very well conceived, investigated and presented. The methodology is mostly made clear (see comments below), substantial interesting findings are presented, and the whole paper may serve as a model for the use of precisely calibrated DEMs for investigating changes in the "Hydrostatic Boundary" at ice shelf-ocean interfaces.

*We thank the reviewer for their supportive comments and address their suggestions below.*

**General Comments**

It is understandable that the (very) recent paper by Eric Rignot (Widespread seawater intrusions…) is not mentioned in this study, probably because it was in review as this paper was being submitted. I recommend that the authors include this paper in their review not simply because it is relevant, but because it could serve to clarify the relationship between the transition zone between grounded and floating ice as detected by InSAR and the Hydrostatic Boundary as measured by DSM analysis. Professor Rignot's paper finds evidence of seawater-induced vertical motion inland beyond the HBs in this paper and the discussion could be quite informative. The adoption of informal names of some sub-glacial features may also be appropriate. From the present high quality of argumentation and discussion I doubt it will take long to add this potentially valuable element.

*This is similar to an apt comment made by Reviewer 2 as well, and we believe we have addressed both reviewers' comments with the additions we've made. Yes, the paper by Eric Rignot et al. was published about a month after this manuscript was submitted, and we immediately started discussing how we could contextualize our results with theirs, so we certainly agree with the reviewers that it is relevant and should be included. While we don't find strong evidence for seawater-induced vertical motion inland of the grounding zone, it is important to note that our DSM analysis is at a much coarser temporal resolution than that of Rignot et al.; the locations of the uplift/subsidence regions identified in this work have been added to several figures and we have added a new paragraph to the Discussion to address this (Section 5.3):*

"One of the channels we observe, ThC7, initiates near where two subglacial drainage channels discharge to the ocean (Rignot et al., 2024). Using differential SAR interferometry, Rignot et al. (2024) observed several circular areas ~4–6 km in diameter with time-varying uplift and subsidence (10–20 cm). These features are located above subglacial topographic depressions that abut km–scale subglacial ridges. The major features are all adjacent to prominent subglacial drainage channels and resemble the isolated HBs we infer inland of the GZ in and around Cavities 1a, 6, 8, and 9 (Figs. 3–5b). Rignot et al. (2024) conclude that the filling and draining of the more inland features is driven by fluctuations in the subglacial water flow through the nearby channels. For the large 'bull's eye' feature just above the grounding line (see Figure 4c in Rignot et al. 2024), however, they speculate that the vertical motion is due to tidally-forced seawater intrusion, which they suggest should cause enhanced basal melting. They do not specify the magnitude of this melt other than to say it should be much lower than 20 m yr$^{-1}$. If this non-steady melting is significantly above the background basal melt rate, we would expect to see a signature in the long-term thinning rates. Instead, the 2020–2023 elevation change data show thinning of 1–2 m yr$^{-1}$ in the area surrounding the downstream feature with minor thickening (<0.5 m yr$^{-1}$) near its centre, providing little or no indication of enhanced melt (Figs. 5f, S7c). We also note that d$H$/d$t$ derived from annual DSM mosaics does not provide the fine temporal resolution (up to sub-daily) over which uplift/subsidence features were observed in this study. We do not observe increased rates of thinning for most of these closed regions, even when they are near the main HB, suggesting that any enhanced melting due to incursion of seawater may not persist long enough to significantly impact the signal on multi-annual timescales for most of the glacier. An alternate hypothesis is that all of the circular features are driven by subglacial water flow rather than seawater intrusion. This hypothesis is supported by a strong gradient in the hydraulic potential between the grounding line and the 'bull's eye' feature, which should drive the water toward – not away from – the ocean (Fig. S6). Seawater intrusion is also problematic because it needs to occur over an area where the predominant flow direction should be seaward to accommodate major subglacial outflows. These features likely fill and drain through exchange of water with the adjacent subglacial channels, similar to how lakes located much farther inland fill and drain (Smith et al., 2017). If this is the case, the pressure boundary condition where these channels meet the ocean should be subject to tidal modulation (10 kPa) sufficient to explain the observed ~10–20 cm uplift/subsidence (1–2 kPa)."

*We have also added a reference to the Rignot et al. paper in the first paragraph of the introduction, as it provides further motivation for the timeliness of our GZ investigations, and adopted the use of "main trunk" rather than "embayment" for the TWIT GZ and other informal feature names for consistency with this and other papers.*

Mostly, the remaining Thwaites ice shelf is referred in this paper to as "TGIS" (Thwaite Glacier Ice Shelf), but I detected some "TEIS" references (Thwaites Eastern Ice Shelf) which is my own preference because it acknowledges the former existence of a western ice shelf. Consistency is obviously required and you (and maybe the Editor) should decide which to use.

*We agree that there was not enough consistency in the use of these acronyms. We have reworked the introduction, where we define the TEIS and TWIT as distinct portions of the shelf, and to define the TGIS as referring to both portions collectively. We have also checked the rest of the manuscript to ensure consistency throughout, making changes as appropriate.*

**Specific comments**

Line 112: which "annual velocity map"?

*We address this together with the following comment (L. 114).*

Line 114: how can you have a "median of two", and how do you define "summer quarters"?

*We have rewritten this description of the methodology; hopefully the reviewer will agree that it provides more clarity:* "The MEaSUREs annual velocity maps obtained for 2011–2015 are variable in their spatial coverage and quality, while the quarterly velocity maps obtained for 2016–2023 have more consistent coverage and better quality. To obtain annual velocity maps from 2011–2023 with more consistent quality, we initially take different approaches to filling data gaps and reducing noise in each dataset: for the 2011-2015 annual velocity maps, we take the average of each annual map and the velocity mosaic at each pixel; for the 2016-2023 quarterly maps, we take the average of each year's Oct–Dec map and Jan–Mar map at each pixel."

Line 137: please expand what you mean by "each independent continuous grounding line"

*We have removed this phraseology as it was unnecessary. The sentence now reads* "We track HBs at the continental grounding line and six pinning points (PP1–6) delineated in the IPY GL ."

Line 148: I admire that you have used 'inclusive' colour scales. It is best not to refer to the (subjectively received) colours in the main text but allow the figures to speak for themselves

*Thank you for appreciating the colour scales, which we put a lot of work into selecting! We have removed this sentence and subsequent references to subjectively received colors in the main text.*

Figure 1: The IPY GL and 2011 GL are apparently in the same colour and the former is probably obscured by the HB sequence. Some adjustments (or removals) are required here.

*We have elected to superimpose the IPY GL over the HBs and change its symbology rather than have it covered by the HBs.*

Line 208: "Remaining artefacts .. are filtered out". Please elucidate.

*We have specified that artefacts may be due to clouds or poorly co-registered strips:* "…extreme values resulting from remaining artefacts from clouds or poorly co-registered strips in the annual mosaics are filtered out."

Line 236: "Several". Why not be precise here?

*Good point. "Several" → "Six".*

Line 261: "TEIS" and "TWIT". I would say these have gained enough currency for general adoption. But then I would.

*This has been addressed in our response to the reviewer's second General Comment above.*

Lines 279 and 286: seven, then eight basal channels?

*Good catch. "Eight" was a typo persisting from a previous version when we were considering a less-convincing feature. There are only seven features that we consider to be basal channels; fixed.*

Line 303: "by the end of the study period". You could help the reader here by giving precise time boundaries.

*We purposefully left this vague because some channels did not experience melting throughout the entire study period, although all did in at least one multi-year epoch. However, the reviewer's point is taken and we have replaced "by the end of the study period" with "within at least one multi-year epoch". As further detail is described in the following sections, we are comfortable with leaving this summary somewhat vague:* "Retreat of the HB exceeding 1 km occurred along all reference channels except for ThC6 and ice–column thinning and melting occurred near all channel intersections with the GZ within at least one multi-year epoch (Figs. 2–6)."

Line 305: I couldn't see how Figure 7 could be used as evidence here.

*Figure 7 shows time-varying velocity and HB position along the reference channels in the vicinity of the GZ, and we did **not** observe any strong patterns among the channels (e.g. Fig. 7 showed no evidence that the velocity increased more with more HB retreat). We have made this sentence more specific to reflect this, and to alert the reader that some individual channels may exhibit a connection between velocity and HB retreat*: "However, no strong relationships emerge between changes in velocity and HB retreat rates along all channels; notable correlations between changes in velocity and changes in HB position along individual channels are described in ensuing sections (Fig. 7)."

Line 405: "was extended .. arbitrarily". Please explain more precisely what you mean.

*We revised this sentence for precision:* "Thus, we manually extended the upstream end of the ThC6 reference channel about 5 km inland of the GZ to show retreat past the IPY GL."

Line 427: "or an error in the manual delineation". This alerted me to the fact that I had missed that a manual step is involved in the method - I had assumed that the process was automated. Perhaps you could expand the methods section to explain this in a nit more detail and discuss the potential errors. Errors in manual steps are rather different from uncertainties in automated processing. I think this sentence needs some more nuance.

*We have added an additional section addressing uncertainties and manual errors in the methods section (Section 3.5), and revised this sentence to provide more context:*

"In 2022, this region only has coverage from one or two strips (Fig. S2), resulting in only two mappings of HB features from which the annual HB was manually delineated. Although some regions are covered by few strips in several years, we are more confident in HB positions that persist or display a pattern over several years, and one year of data does not provide sufficient evidence to conclude that this entire region was ungrounded in 2022."

Line 442: extra brackets.

*Good catch. Fixed.*

Line 520: "volume of basal melt .. 3.5Gt". Please give a time period as well as an area. To claim this as "ample" requires some more data or argumentation.

*We have added additional details based on closer comparison of our results with the cited paper:*

"The basal meltwater volume has been estimated at 3.5 Gt a$^{-1}$ for the 189,000 km$^2$ Thwaites Glacier drainage basin, with most of the melt occurring within about 50 km inland of the GZ (Joughin et al., 2009). Our study area extends from ~10-100 km inland of the GZ, so ample subglacial water is available, and may discharge…"

Line 529: Figure S8a would need some more annotation to support this statement.

*We have labeled the pinning points in Fig. S8a and marked the highest point of the bathymetry in this region with a '*' marker.*

*Great work.*

Adrian Luckman, 22nd May 2024

*We thank Adrian wholeheartedly for his helpful and constructive comments, which we believe have greatly improved the readability and impact of this paper.*

---

## Author Comment (AC2)

**Response to Reviewer 2**

This study present new observation of grounding line migration and basal melting under the floating section of the Thwaites glacier of the West Antarctica Ice Sheet. The study focuses particularly on the link between the presence of basal channels, basal melting, and grounding line migration. To achieve this, the authors analyse an extensive dataset of high-resolution digital elevation models, providing a unique lens into the intricate processes taking place at grounding lines of a rapidly melting sector of Antarctica.

The research is very topical and the findings provide novel insights into the potential role of channels in modulating both the melting under ice shelves as well as grounding retreat. The manuscript is very well written and illustrated, the method is very well documented, the results and discussion are informative and provide a balanced view of findings and limitations.

*We thank the reviewer for their supportive comments and address their specific comments below.*

**Specific comments:**

Line 35-38: It is unclear here what the relative role of subglacial channelization and meltwater plumes is in creating these ice shelves basal channels.

*We agree that this phrasing was unclear. We have rewritten these lines to make it clear that it is not fully known how any given channel is formed, although if subglacial channelization is present, it can enhance freshwater plumes:*

"They often represent advected extensions of inverted troughs initiated by subglacial channelization beneath the grounded ice or incised by undulations in the bed (e.g. Le Brocq et al., 2013; Alley et al., 2016; Drews et al., 2017). Where subglacial channelization is present, the input of fresh subglacial meltwater may contribute to the growth of a buoyant meltwater plume that can entrain warm ocean water as it travels along the basal channel (Jenkins, 2011). However, it remains difficult to attribute the formation mechanism to any given channel, particularly if its surface expression does not intersect the grounding line (e.g. Alley et al., 2016; Chartrand & Howat 2020)."

L83: SAR-In -> SARIn

*Fixed.*

L122: Could you say more about the 07-09 GL datasets? Is this the 2011 grounding line in https://agupubs.onlinelibrary.wiley.com/doi/full/10.1002/2014GL060140 ? Which would then tie in nicely with the start of your HB record in 2011.

*We have added additional details about the derivation of the 07-09 IPY GL and why we selected it, resulting in a rewritten paragraph for Section 2.4:*

"The grounding line from the MEaSUREs Antarctic Boundaries for the 2007–2009 International Polar Year (IPY) from Satellite Radar, Version 2 dataset (Mouginot et al., 2017b) is used as a reference grounding line from which to measure changes in the grounding zone position and is henceforth termed 07–09 IPY GL or simply IPY GL. This dataset provides a complete and continuous grounding line derived from a variety of satellite platforms. Additional historical grounding lines are obtained for a long term visual comparison (Fig. 1) from the MEaSUREs Antarctic Grounding Line from Differential Satellite Radar Interferometry, Version 2 for 7 February 1992 to 17 December 2014 (Rignot et al., 2016); however these are not used for analyses."

*The DOI for each GL dataset is provided with its respective reference in the reference list and in the Code and Data Availability section. We chose not to include the GL product in the mentioned publication because it is not continuous and represents a different grounding line proxy than our HBs, as described in the newly added* Section 3.5 Uncertainties and sources of error.

L203: Should it say "the velocity divergence is computed at each time step prior to the DSM being flow–shifted, …"?

*Yes, good catch and accurate inference. Fixed.*

L206: What are the implications of having the SMB data only up to 2016 when the melt rate is calculated up to 2023? What would the impact of large anomalies e.g. https://www.nature.com/articles/s41467-023-36990-3 be?

*We have added a sentence within the new Section 3.5 to explain that were are not concerned with short-term, basin-wide SMB anomalies because we are interested in spatial variability over several years:*

"We note that $M_s$ is derived from a temporal average of RACMO model output from only part of our study period, which may omit the impact of anomalous precipitation events on our estimates of $M_b$. However, as we are interested in the spatial variability of GZ change over several years, we do not expect the omission of short–term, regional events to significantly impact our results as they will be partially captured in DSM surface heights."

Figure 3: (g) and (h) labels are only partially visible.

*Fixed.*

L255: Not sure what you mean here? What other than melting would cause basal mass change? Please clarify.

*We agree that this phrasing was unclear. We have revised this sentence to:* "rates of basal mass change are predominantly negative, indicating melting, but are…".

L256: I suggest caution in how you present "thinning" over ice shelves, this is Lagrangian change in elevation – "thinning" might create confusion with the reader. I would suggest using a different term. This applies to many sections of the manuscript, where elevation and thickness change over floating ice is mentioned.

*We thank the reviewer for this suggestion, and address it together with their comment on L457. We have rewritten the first paragraph of Section 3.3 to more clearly distinguish the meaning of "thinning" in a Lagrangian frame versus an Eulerian frame, and we now articulate what can lead to ice shelf thinning in a Lagrangian reference frame:*

**"3.3 Estimating rates of change**

Time–evolving rates of change are estimated from the annual DSM mosaics within four epochs: 2011–2015, 2016–2019, 2020–2023, and the entire study period from 2011–2023. Within each epoch, rates of change are calculated from all combinations of annual mosaics such that the relevant quantity derived from the earlier mosaic in each combination is subtracted from the later mosaic and divided by the time elapsed between the mosaics. The Eulerian reference frame (fixed–coordinate system, denoted by $dQ/dt$, where $Q$ is the quantity in question) is used over grounded ice, to prevent slope–induced errors, and the Lagrangian reference frame (coordinate system moves with ice flow, denoted by $DQ/Dt$) is used over floating ice, where height variability is dominated by horizontal advection. The strip–derived annual HB from each year is used to delineate the extent of floating and grounded ice for each annual mosaic. For grounded ice, we calculate the Eulerian rate of thickness change ($dH/dt$), where grounded ice thickness, $H$, is simply the DSM–derived surface height minus the BedMachine bed height. For floating ice, we calculate Lagrangian rates of ice–column surface height change ($Dh/Dt$), thickness change ($DH_E/Dt$), where flotation thickness $H_E$ is derived from annual DSM mosaic freeboard heights using Eq. 1, and basal mass loss or gain ($M_b$). For Lagrangian calculations, the mosaics are flow–shifted to a common date using the smoothed annual surface velocity maps (Section 2.3) following the approach of Shean et al. (2019) and Chartrand & Howat (2020). The mosaics are flow–shifted to 1 January of the earliest full year in each epoch (e.g., 1 January 2011 for the 2011–2015 epoch and the full study period). Lagrangian ice–column thinning can occur as a result of stretching as the ice accelerates (dynamic thinning) or as a result of surficial or basal ablation, although these mechanisms cannot be attributed by a calculation of $DH_E/Dt$, which only reflects how the surface height changes as the column advects due to the hydrostatic assumption."

*We have also added a sentence at the location of this comment to reiterate the meanings of "thinning"* ("For ice shelf thickness changes in the Lagrangian frame ($DH_E/Dt$), thinning refers to change in the same column of ice as it advects with flow, rather than thinning at a fixed coordinate (Eulerian Frame), which we refer to on grounded ice."*), and we have changed all other references to Lagrangian changes on the ice shelf to "ice-column thinning"/elevation change/etc. or similar.*

L263: Meaning that apparent refreezing is an artefact of the floating assumption? Or does the transient grounding leads to real refreezing somehow? Please clarify.

*We agree that this was unclear; this sentence had gone through a lot of word-smithing in the past and this comment helped us see it with fresh eyes. The reviewer correctly inferred that the apparent refreezing is an artefact of the floating assumption; the sentence has been revised to reflect this:* "We expect that the apparent positive $M_b$ in the downstream portion of the TWIT may be an artefact of hydrostatic disequilibrium due to transient grounding, as evidenced…"

L276: Is it "everywhere"? In several sectors (cavities 6, 7, 8 &9), IPY GL appear to be inland of the HB position in ~ 2011 and 2012. Given the importance of this sector it is probably worth discussing and providing potential explanation for this.

*We hope that the rewording of Section 2.4 and the addition of Section 3.5 has helped make it clear that the IPY GL reflects the furthest inland position where the ice experiences tidal displacement, while the HBs reflect the furthest inland position where the ice is in hydrostatic equilibrium, so they should not be directly compared as they can differ by several km (as described by Helen Fricker and others (Fig. 2 in particular) DOI:10.1017/S095410200999023X). We have also added additional clarifying details to this paragraph to make this distinction: "...*stagnated relative to its early positions everywhere by 2023, including on pinning points, with significant variability in the rates of retreat, including some small and temporary areas of advance. In Cavities 6–9, early HBs appear seaward of the 07–09 IPY GL, likely due to the differences in mapping method, but by 2023 the HB had also retreated or stagnated relative to the IPY GL everywhere."

L450: I am curious whether you observe the pinning point evolution at TWIT described in: https://doi.org/10.1029/2023GL103088 from the unfiltered figure 5 it appears so but it would be worth a mention, and why this pinning point may or may not be more robust than the other unfiltered HB features.

*We thank the reviewer for making this connection between Fig. 5b and the ice rumple tracked by Holland et al.. We have added a reference to this paper here:* "The IPY GL did not contain a pinning point in the main trunk of the TWIT, although Holland et al. (2023) track the evolution of an ice rumple near the centre of the main trunk, which disappeared between 2011–2022." *and additional details about our observations in the last paragraph of this section:* We also map an isolated HB near this ice rumple (labeled "HR" for "Holland Rumple" in Figure 5b) which disappears by 2014." *and the last paragraph of Section 5.2:* "...we are less confident in the existence of additional pinning points within the TGIS, with the exception of the "HR" ice rumple which was independently mapped by Holland et al. (2023) using similar methods (Fig. 5b)."

L466: It would be good for the discussion to reflect on the implication of the findings in light of the recent publications (e.g. https://www.pnas.org/doi/full/10.1073/pnas.2404766121 but also others) on ocean intrusion within the grounding zone, possibly by expanding some of the related discussion in section 5.2. Your comments on the absence of elevation thinning in these sectors for example seem particularly relevant.

*This is similar to an apt comment made by Reviewer 1 as well, and we believe we have addressed both reviewers' comments with the additions we've made. The paper by Eric*

*Rignot et al. was published about a month after this manuscript was submitted, and we immediately started discussing how we could contextualize our results with theirs, so we certainly agree with the reviewers that it is relevant and should be included. While we don't find strong evidence for seawater-induced vertical motion inland of the grounding zone, it is important to note that our DSM analysis is at a much coarser temporal resolution than that of Rignot et al.; the locations of the uplift/subsidence regions identified in this work have been added to several figures and we have added a new paragraph to the Discussion to address this (Section 5.3):*

"One of the channels we observe, ThC7, initiates near where two subglacial drainage channels discharge to the ocean (Rignot et al., 2024). Using differential SAR interferometry, Rignot et al. (2024) observed several circular areas ~4–6 km in diameter with time-varying uplift and subsidence (10–20 cm). These features are located above subglacial topographic depressions that abut km–scale subglacial ridges. The major features are all adjacent to prominent subglacial drainage channels and resemble the isolated HBs we infer inland of the GZ in and around Cavities 1a, 6, 8, and 9 (Figs. 3–5b). Rignot et al. (2024) conclude that the filling and draining of the more inland features is driven by fluctuations in the subglacial water flow through the nearby channels. For the large 'bull's eye' feature just above the grounding line (see Figure 4c in Rignot et al. 2024), however, they speculate that the vertical motion is due to tidally-forced seawater intrusion, which they suggest should cause enhanced basal melting. They do not specify the magnitude of this melt other than to say it should be much lower than 20 m yr$^{-1}$. If this non-steady melting is significantly above the background basal melt rate, we would expect to see a signature in the long-term thinning rates. Instead, the 2020–2023 elevation change data show thinning of 1–2 m yr$^{-1}$ in the area surrounding the downstream feature with minor thickening (<0.5 m yr$^{-1}$) near its centre, providing little or no indication of enhanced melt (Figs. 5f, S7c). We also note that d$H$/d$t$ derived from annual DSM mosaics does not provide the fine temporal resolution (up to sub-daily) over which uplift/subsidence features were observed in this study. We do not observe increased rates of thinning for most of these closed regions, even when they are near the main HB, suggesting that any enhanced melting due to incursion of seawater may not persist long enough to significantly impact the signal on multi-annual timescales for most of the glacier. An alternate hypothesis is that all of the circular features are driven by subglacial water flow rather than seawater intrusion. This hypothesis is supported by a strong gradient in the hydraulic potential between the grounding line and the 'bull's eye' feature, which should drive the water toward – not away from – the ocean (Fig. S6). Seawater intrusion is also problematic because it needs to occur over an area where the predominant flow direction should be seaward to accommodate major subglacial outflows. These features likely fill and drain through exchange of water with the adjacent subglacial channels, similar to how lakes located much farther inland fill and drain (Smith et al., 2017). If this is the case, the pressure boundary condition where these channels meet the ocean should be subject to tidal modulation (10 kPa) sufficient to explain the observed ~10–20 cm uplift/subsidence (1–2 kPa)."

*We have also added a reference to the Rignot et al. paper in the first paragraph of the introduction, as it provides further motivation for the timeliness of our GZ investigations.*

L475: "Rapid thinning" in a Lagrangian sense has a different meaning than the general understanding of ice shelf thinning. I would urge caution. Somewhere in the manuscript it would be good to articulate what can lead to ice shelf thinning in a Lagrangian reference frame, and in the discussion to address the plausibility of various processes.

*We hope that our response to the reviewer's comment on L256 sufficiently addresses this comment.*

L553: Could you discuss the implication of this magnitude of error on the smooth annual HB? Could this explain some of the discrepancy with the IPY GL in the TWIT sector (fig. 5)?

*We hope that the addition of* Section 3.5 Uncertainties and sources of error *and our response to the reviewer's comment on L276 have helped to sufficiently address this comment.*

*We thank the reviewer wholeheartedly for their helpful and constructive comments, which we believe have greatly improved the readability and impact of this paper.*

---

## Author Response (AR2)

**Author's response to Technical Corrections**

l 106 accounting for firn density (missing reference which model was used)

*We have specified the firn model used.*

Reference Zinck et al. 2023 is by now peer reviewed and out of the discussion phase. Unless you are refering to content exclusively available in the Discussion I suggest you cite the peer-reviewed version.

*Thanks for catching this. Peer-reviewed version is now cited.*

Reference to Topo-Toolbox appears incorrect. Looking briefly at their website it appears you should credit:

"Schwanghart, W., Scherler, D. (2014): TopoToolbox 2 – MATLAB-based software for topographic analysis and modeling in Earth surface sciences. Earth Surface Dynamics, 2, 1-7. [DOI: 10.5194/esurf-2-1-2014]"

*Thanks for catching this. Corrected.*

Figure 2 has no scalebar

*We have added a scalebar and north arrow to the middle panel, consistent with Figs. 3-5.*

Figure 4 b needs label on colorbar (year)

*We have added a label to all HB colorbars.*